# BRI1 and BAK1 interact with G proteins and regulate sugar-responsive growth and development in *Arabidopsis*

Yuancheng Peng [1,2], Liangliang Chen[1,3], Shengjun Li[1], Yueying Zhang[1,3], Ran Xu[1], Zupei Liu[1,3], Wuxia Liu[1,3], Jingjing Kong[2], Xiahe Huang[4], Yingchun Wang[4], Beijiu Cheng[2], Leiying Zheng[5] & Yunhai Li [1,3]

Sugars function as signal molecules to regulate growth, development, and gene expression in plants, yeasts, and animals. A coordination of sugar availability with phytohormone signals is crucial for plant growth and development. The molecular link between sugar availability and hormone-dependent plant growth are largely unknown. Here we report that BRI1 and BAK1 are involved in sugar-responsive growth and development. Glucose influences the physical interactions and phosphorylations of BRI1 and BAK1 in a concentration-dependent manner. BRI1 and BAK1 physically interact with G proteins that are essential for mediating sugar signaling. Biochemical data show that BRI1 can phosphorylate G protein β subunit and γ subunits, and BAK1 can phosphorylate G protein γ subunits. Genetic analyses suggest that BRI1 and BAK1 function in a common pathway with G-protein subunits to regulate sugar responses. Thus, our findings reveal an important genetic and molecular mechanism by which BR receptors associate with G proteins to regulate sugar-responsive growth and development.

[1] State Key Laboratory of Plant Cell and Chromosome Engineering, CAS Centre for Excellence in Molecular Plant Biology, Institute of Genetics and Developmental Biology, Chinese Academy of Sciences, Beijing 100101, China. [2] School of Life Sciences, Anhui Agricultural University, Hefei 230036, China. [3] University of Chinese Academy of Sciences, Beijing 100039, China. [4] State Key Laboratory of Molecular Developmental Biology, Institute of Genetics and Developmental Biology, Chinese Academy of Sciences, Beijing 100101, China. [5] Key Laboratory of Plant Molecular Physiology, Institute of Botany, Chinese Academy of Sciences, Beijing 100093, China. These authors contributed equally: Yuancheng Peng, Liangliang Chen, Shengjun Li. Correspondence and requests for materials should be addressed to B.C. (email: chengbeijiu@126.com) or to L.Z. (email: lyzheng@ibcas.ac.cn) or to Y.L. (email: yhli@genetics.ac.cn)

Sugars not only play crucial roles as energy sources and carbon skeleton supply but also act as signal molecules that regulate a variety of growth and developmental processes in yeasts, animals, and plants[1–3]. Several sugar sensing and signaling mechanisms are evolutionarily conserved in yeasts, animals, and plants, such as G-protein signaling and TOR signaling[1–3]. Unlike yeasts and animals, plants are autotrophic organisms that produce sugars by photosynthesis. Sugars are transported from their source of production to regions of high respiratory and storage demand in plants. Thus, plants might possess additional regulatory mechanisms by which plants sense the status of carbon to regulate their growth and development.

Several regulatory pathways involved in plant sugar sensing and signaling have been identified by their conservation among yeasts, animals, and plants. *Arabidopsis* hexokinase (HXK1) functions as a glucose sensor[4]. Plant SNF1-related kinase 1 (SnRK1) proteins, which are homologs of AMP-activated protein kinases (AMPK) in mammals and Snf1 sucrose non-fermenting 1 (Snf1) proteins in yeasts, play crucial roles in sugar metabolism and sugar signaling[5–7]. TARGET OF RAPAMYCIN (TOR) acts antagonistically to the starvation-induced AMPK/Snf1 kinases in animals and yeasts[8]. The plant TOR complex has been shown to link photosynthesis-driven glucose nutrient status with growth processes[9]. G-protein signaling has also been known to mediate sugar responses in yeasts and *Arabidopsis*[10–13]. G-protein-coupled pathways transmit a signal, via a membrane receptor and G-protein subunits (Gα, Gβ, and Gγ), to downstream effectors. In *Arabidopsis*, sugar-induced G-protein activation is mediated by the regulator of G-protein signaling 1 (RGS1), a putative receptor or co-receptor of glucose[13,14]. In *Arabidopsis*, mutations in *RGS1* result in sugar-insensitive growth, while mutations in G-protein subunits cause sugar-hypersensitive phenotypes[12,15–17].

Crosstalks between sugar and several phytohormone signaling pathways have been described in plants[18–23]. Several ABA biosynthetic mutants and ABA response mutants exhibit the reduced responses of seedlings to high levels of glucose or sucrose[18–20]. Mutations in genes involved in ethylene signaling pathways cause altered sugar responses[22,23]. Recent studies reveal an overlap between brassinosteroid (BR) signaling and sugar promotion of hypocotyl elongation[24–27]. BRs are perceived by the BR receptor BRASSINOSTEROID INSENSITIVE 1 (BRI1)[28]. BRI1-associated kinase 1 (BAK1) interacts with BRI1 to form the receptor complex, initiating phosphorylation cascades, and eventually regulating the expression of downstream target genes[29–31]. However, how BRI1 and BAK1 regulate sugar signaling is almost unknown.

In this study, we demonstrate that BRI1 and BAK1 physically associate with G-protein subunits. Biochemical data show that BRI1 and BAK1 phosphorylate G-protein subunits. Genetic analyses suggest that BRI1 and BAK1 function in a common pathway with G-protein subunits to control sugar-responsive growth and development. Thus, our findings define an important genetic and molecular mechanism by which BR receptors interact with G-protein subunits to regulate sugar-responsive growth and development.

## Results

**BRI1 and BAK1 play key roles in sugar responses.** Dark-grown *Arabidopsis* seedlings develop leaf-like organs on vertical Petri dishes with even very low concentrations of sugars, suggesting that the dark development phenotype of *Arabidopsis* seedlings is a sensitive indicator of the effects of sugars on plant growth and development[32–34]. To understand how brassinosteroids (BRs) influence sugar responses, we investigated the dark development phenotype of 19-d-old Col-0, *bri1-301*, and *bak1-4* seedlings. As shown in Fig. 1a, the development of dark-grown *Arabidopsis* seedlings grown on MS medium with glucose was classified into four different stages[32–34]. At the stage 1, seedlings did not show

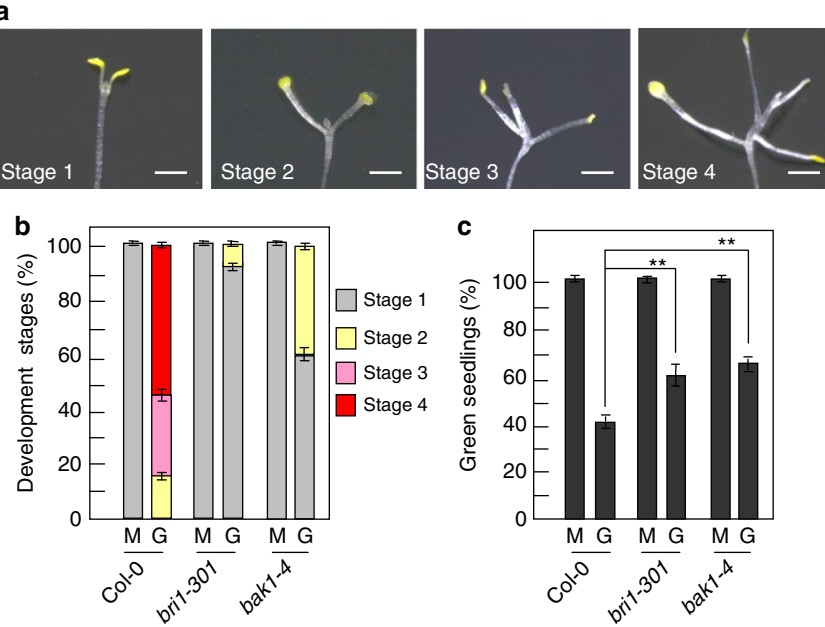

**Fig. 1** BRI1 and BAK1 play key roles in sugar responses. **a** Different stages of dark-grown wild-type seedlings used for scoring sugar responses. **b** Comparison of developmental stages between Col-0, *bri1-301*, and *bak1-4*. Seedlings were grown vertically on medium supplemented with 1% glucose (G) or 1% mannitol (M) in the dark for 19 days (*n* ≥ 83). **c** The percentage of Col-0, *bri1-301*, and *bak1-4* seedlings with green cotyledons. Seedlings were grown on medium supplemented with 6% glucose (G) or 6% mannitol (M) under constant light condition for 10 days (*n* ≥ 89). Values in **b**, **c** are given as mean ± SD. **P < 0.01 compared with the wild type by Student's *t*-test. Scale bars, 2 mm (**a**)

the expansion of cotyledons. At the stage 2, seedlings had fully expanded cotyledons and the first pair of leaves. At the stage 3, seedlings had developed first pair of true leaves but internode did not appear. At the stage 4, seedlings had fully developed first pair of leaves and a clear internode, and more leaves started to form. The glucose-induced dark development of *Arabidopsis* seedlings was not a result of an osmotic effect, because seedlings never developed beyond the expansion of cotyledons on medium containing 1% mannitol (Supplementary Fig. 1a), consistent with previous reports[32–34]. As shown in Fig. 1b and Supplementary Fig. 1b, most wild-type seedlings grown on MS medium with 1% glucose had developed to the stage 3 or the stage 4. By contrast, most *bri1-301* and *bak1-4* seedlings only developed to the stage 1. These results indicate that dark-grown *bri1-301* and *bak1-4* seedlings are insensitive to exogenous glucose based on the phenotypes of leaf-like structure formation, cotyledon expansion, and internode elongation. In dark-grown Col-0 seedlings, hypocotyl length increased in response to low concentrations of glucose, and elongation was progressively inhibited at high concentrations of glucose (Supplementary Fig. 2)[33,34]. Hypocotyl elongation of dark-grown *bri1-301* and *bak1-4* seedlings was insensitive to exogenous glucose (Supplementary Fig. 2). These results show that BRI1 and BAK1 influence sugar-regulated hypocotyl elongation.

During early seedling morphogenesis, high glucose inhibited seedlings to form green cotyledons and leaves under light condition, resulting in the developmental arrest[4,20]. As shown in Fig. 1c, most wild-type seedlings grown on medium with 6% glucose under constant light condition did not develop green cotyledons. In contrast, most *bri1-301* and *bak1-4* seedlings had green cotyledons, indicating that light-grown *bri1-301* and *bak1-4* are insensitive to high-glucose-induced developmental arrest (Fig. 1c and Supplementary Fig. 1c). Taken together, these results demonstrate that BRI1 and BAK1 play key roles in regulating sugar-responsive growth and development under both dark and light conditions.

**Glucose influences the interactions between BRI1 and BAK1**. Ligand-induced BRI1 associates with its co-receptor BAK1 to activate BR signaling[30,31,35,36]. We therefore asked whether glucose could regulate the physical interactions between BRI1 and BAK1 in *Arabidopsis*. The *pBRI1:BRI1-GFP;35S:BAK1-HA* seedlings were treated with different concentrations of glucose for 4 or 24 h. The interactions between BRI1-GFP and BAK1-HA were progressively increased in response to 1 and 2% glucose, and then gradually decreased in response to 4 and 6% glucose (Fig. 2a, b). To confirm this, we examined the effects of glucose on the interactions of BRI1 and BAK1 using the native antibodies of BRI1 and BAK1. Similar results were observed in wild-type seedlings treated with glucose (Supplementary Fig. 3). The interactions between BRI1-GFP and BAK1-HA in seedlings treated with 1% glucose for 24 h were much stronger than those in 4 h-treated seedlings. By contrast, mannitol did not affect the physical interactions between BRI1-GFP and BAK1-HA (Supplementary Fig. 4). Thus, these results show that glucose influences the physical interactions between BRI1 and BAK1 in a concentration-dependent manner.

The interactions between BRI1 and BAK1 have been known to cause their reciprocal transphosphorylation[35,37]. We then asked whether glucose could modulate the phosphorylation levels of BRI1 and BAK1. Total proteins were isolated from *pBRI1:BRI1-GFP* seedlings treated with different concentrations of glucose and incubated with GFP-Trap-A agarose beads to immunoprecipitate BRI1-GFP. Immunoprecipitated proteins were detected with anti-GFP and anti-phosphoserine (anti-pSer) antibodies,

respectively. As shown in Fig. 2c, d, the phosphorylation levels of BRI1-GFP were progressively increased in response to 1 and 2% glucose, and then gradually reduced at glucose concentrations between 4 and 6%. We also treated *pBAK1:BAK1-GFP* seedlings with different concentrations of glucose. Considering that the anti-phosphothreonine (anti-pThr) antibody was often used to detect the phosphorylated BAK1[37–39], we therefore used the anti-pThr antibody to examine the phosphorylated BAK1-GFP. As shown in Fig. 2e, f, glucose affects the phosphorylation levels of BAK1 in a glucose concentration-dependent manner. Similar results were observed when we used the native antibodies of BRI1 and BAK1 to examine their phosphorylation levels, respectively (Supplementary Fig. 5). By contrast, mannitol did not affect the phosphorylation levels of BRI1-GFP and BAK1-HA (Supplementary Fig. 6). Thus, these results indicate that glucose modulates the phosphorylation levels of BRI1 and BAK1 in a concentration-dependent manner.

BR signaling is mainly activated by the plasma-membrane-localized BRI1[40–42]. BR signaling is enhanced when BRI1 internalization is inhibited, and BRI1 endocytosis is mainly required for BR signal attenuation[40–42]. We asked whether glucose could affect the plasma membrane localizations of BRI1 and BAK1 in plants. The plasma membrane localizations of BRI1-GFP and BAK1-GFP were not strongly affected in response to glucose from 0 to 2% (Supplementary Fig. 7). However, the plasma membrane localizations of BRI1-GFP and BAK1-GFP were significantly decreased in response to glucose from 4 to 6% (Supplementary Fig. 7), consistent with the notion that high concentrations of glucose inhibit plant growth. Thus, these results indicate that glucose modulates the plasma membrane localizations of BRI1 and BAK1 in a concentration-dependent manner.

**G-proteins act genetically with BRI1 and BAK1**. To understand how BRI1 and BAK1 regulate sugar responses, we sought to identify the genetic points of convergence of BR receptors and sugar signaling pathways. A previous study showed that flg22 induces the interaction between BAK1 and the *Arabidopsis* immune receptor FLS2, which activates G-protein signaling and modulates immune response[43]. Interestingly, glucose influences the physical interactions between BRI1 and BAK1 (Fig. 2a, b), and G-protein signaling has been known to regulate sugar responses in *Arabidopsis*[10–13]. We therefore asked whether crosstalk between BR and sugar could act through the G-protein pathway. We investigated the combined effects of Brassinolide (BL), Brassinazole (BRZ, a chemical inhibitor of BR biosynthesis) and sugars on G-protein mutants. When seedlings were grown on medium without sugars, BRZ and BL did not affect dark development (Supplementary Fig. 8a), supporting the essential role of sugars in dark development. When seedlings were grown on medium with 1% glucose, BRZ strongly inhibited sugar-responsive dark development (Supplementary Fig. 8b). There was no obviously difference between the BL-treated seedlings and the control when seedlings were grown on medium with 1% glucose (Supplementary Fig. 8b). However, the inhibitory effects of BRZ on dark development could be rescued by BL (Supplementary Fig. 8b). Similar results were also observed when seedlings were grown on medium with sucrose, BRZ and/or BL (Supplementary Fig. 8c). By contrast, the dark development of *gpa1-101*, *agb1-2*, and *agg3-3* was insensitive to the inhibition of BRZ compared with that of the wild type (Supplementary Fig. 9). Interestingly, responses of wild type and G-protein mutant hypocotyls to the combination of glucose, BRZ and BL were not obviously different (Supplementary Fig. 10), consistent with a previous report[24]. Under the light condition, *gpa1-101*, *agb1-2*, and *agg3-3* seedlings were less sensitive to BRZ when seedlings

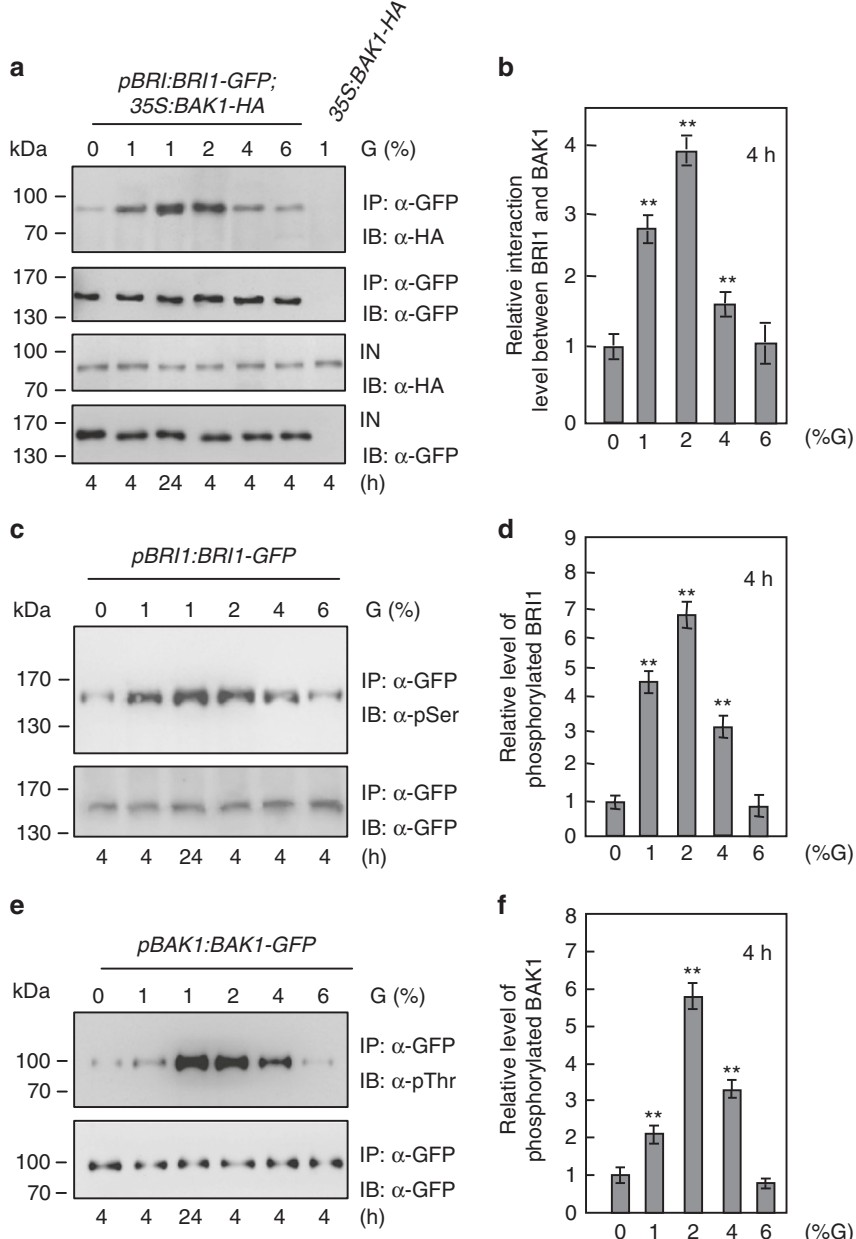

**Fig. 2** Glucose influences the interactions between BRI1 and BAK1. **a** Glucose regulates the physical interactions between BRI1 and BAK1. *pBRI1:BRI1-GFP;35S: BAK1-HA* or *35S:BAK1-HA* seedlings were grown in the light for 5 days, then incubated in darkness for 4 days, and treated with different concentrations of glucose (G) for 4 or 24 h. IN input; IP immunoprecipitation; IB immunoblot. **b** Quantification of BRI1-GFP and BAK1-HA associations in Arabidopsis seedlings treated with different concentrations of glucose (G) for 4 h from **a**. The ratio value of immunoprecipitated BAK1-HA to input BAK1-HA in response to 0% glucose was set at 1. Values for glucose treatments are given as mean ± SD ($n = 3$). **$P < 0.01$ compared with the value for 0% glucose treatment by Student's *t*-test. **c** Glucose influences the phosphorylation levels of BRI1 in Arabidopsis. *pBRI1:BRI1-GFP* seedlings were grown in the light for 5 days, then incubated in darkness for 4 days, and treated with different concentrations of glucose (G) for 4 or 24 h. IN input; IP immunoprecipitation; IB immunoblot. **d** Quantification of BRI1-GFP phosphorylation in Arabidopsis seedlings treated with different concentrations of glucose for 4 h from **c**. The ratio value of phosphorylated BRI1-GFP to immunoprecipitated BRI1-GFP in response to 0% glucose was set at 1. Values for glucose treatments are given as mean ± SD ($n = 3$). **$P < 0.01$ compared with the value for 0% glucose treatment by Student's *t*-test. **e** Glucose influences the phosphorylation levels of BAK1 in Arabidopsis. *pBAK1:BAK1-GFP* seedlings were grown in the light for 5 days, then incubated in darkness for 4 days, and treated with different concentrations of glucose (G) for 4 or 24 h. IN, input; IP immunoprecipitation; IB immunoblot. **f** Quantification of BAK1-GFP phosphorylation in Arabidopsis seedlings treated with different concentrations of glucose for 4 h from **e**. The ratio value of phosphorylated BAK1-GFP to immunoprecipitated BAK1-GFP was set at 1. Values for glucose treatments are given as mean ± SD ($n = 3$). **$P < 0.01$ compared with the value for 0% glucose treatment by Student's *t*-test

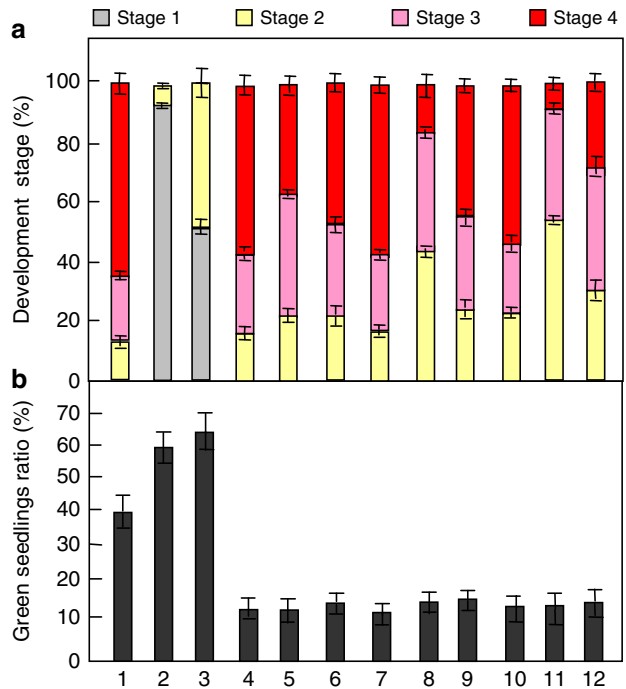

**Fig. 3** G-proteins act genetically with BRI1 and BAK1. **a** Comparison of development stages of Col-0 (1), *bri1-301* (2), *bak1-4* (3), *agb1-2* (4), *agb1-2 bri1-301*(5), *agb1-2 bak1-4* (6), *gpa1-101*(7), *gpa1-101 bri1-301* (8), *gpa1-101 bak1-4* (9), *agg3-3* (10),*agg3-3 bri1-301*, (11), and *agg3-3 bak1-4* (12). Seedlings were grown vertically on medium supplemented with 1% glucose in the dark for 19 days (*n* ≥ 85). **b** The percentage of the indicated seedlings with green cotyledons. Seedlings were grown on medium supplemented with 6% glucose under constant light condition for 16 days (*n* ≥ 72). Values in a.b are given as mean ± SD

were grown on medium with 6% glucose (Supplementary Fig. 11). These results suggest that the regulation of hypocotyl elongation mediated by BR and sugar is not dependent on G proteins, consistent with previous studies[24,25]. However, the regulation of dark developmental stages and growth arrest mediated by BR and sugar is dependent on G proteins.

It has been reported that the *gpa1* and *agb1* mutants were hypersensitive to sugars[12,16,44]. Therefore, we firstly tested genetic interactions of *BRI1* and *BAK1* with *GPA1* and *AGB1*, respectively. The *gpa1-101* and *agb1-2* single-mutant seedlings grown on medium with 1% glucose for 19 days developed to the stage 3 or the stage 4 in the dark, similar to those observed in wild-type seedlings (Fig. 3a). Importantly, most *bri1-301 gpa1-101* and *bri1-301 agb1-2* double mutants had developed to the stage 3 or the stage 4 in the darkness, although most *bri1-301* seedlings only developed to the stage 1, indicating that *GPA1* and *AGB1* act in a common pathway with *BRI1* to regulate dark-grown processes including leaf-like structure formation, cotyledon expansion and internode elongation. Similarly, *GPA1* and *AGB1* act in a common pathway with *BAK1* to regulate these dark-grown processes (Fig. 3a). The green cotyledon percentages of *gpa1-101* and *agb1-2* seedlings grown on medium with 6% glucose under constant light were lower than that of wild-type seedlings (Fig. 3b), indicating that *gpa1-101* and *agb1-2* are sensitive to high glucose. The green cotyledon percentage of *bri1-301* seedlings was higher than that of wild-type seedlings, while the green cotyledon percentages of *bri1-301 gpa1-101* and *bri1-301 agb1-2* double mutant seedlings were similar to those of *gpa1-101* and *agb1-2* single mutant seedlings, respectively. These

results suggest that *gpa1-101* and *agb1-2* are epistatic to *bri1-301* with respect to high-glucose-induced developmental arrest. Similarly, *gpa1-101* and *agb1-2* are epistatic to *bak1-4* in high-glucose-induced developmental arrest (Fig. 3b). These genetic analyses suggest that BRI1 and BAK1 function in a common genetic pathway with GPA1 and AGB1 to control sugar-responsive growth and development.

Arabidopsis Gγ subunits (AGG1, AGG2, and AGG3) interact with AGB1 to regulate plant growth and development[45–47]. The *agg3* mutants showed similar growth phenotypes to *agb1*, such as round and small leaves and small flowers[45,46,48,49]. However, neither *agg1* and *agg2* single mutants or *agg1 agg2* double mutants exhibited the small organ phenotype[48,49], suggesting that AGG3 plays major role in plant growth and development. Therefore, we focused on testing the genetic interactions of BRI1 and BAK1 with AGG3 in sugar-responsive growth. As shown in Fig. 3a, *agg3-3* partially suppressed the dark development phenotypes (e.g., leaf-like structure formation, cotyledon expansion, and internode elongation) of *bri1-301* and *bak1-4*, although *agg3-3* single mutant showed similar dark development phenotypes to the wild type. The green cotyledon percentage of *agg3-3* seedlings grown on medium with 6% glucose under constant light was lower than that of wild-type seedlings (Fig. 3b), revealing that *agg3-3* is sensitive to high glucose. The percentages of *bri1-301 agg3-3* and *bak1-4 agg3-3* double mutant seedlings with green cotyledons were similar to that of *agg3-3* single-mutant seedlings with green cotyledons. These results suggest that *agg3-3* is epistatic to *bri1-301* and *bak1-4* with respect to dark development and high-glucose-induced developmental arrest.

Arabidopsis RGS1, which controls G-protein signaling, is required for normal glucose sensing, and *rgs1* mutants are insensitive to high-glucose-induced developmental arrest[14,15]. In addition, it has been proposed that plant receptor-like kinases, such as LRR receptor-like kinase, FLS2, and BAK1 might phosphorylate RGS1 in flg22-induced defense responses[50]. We therefore asked whether there are any genetic interactions of RGS1 with BRI1 and BAK1 in high-glucose-induced developmental arrest. As shown in Supplementary Fig. 12 and 13, the green cotyledon percentages of *bri1-301 rgs1-2* and *bak1-4 rgs1-2* double-mutant seedlings grown on medium with 6% glucose under constant light were significantly higher than those of their respective single-mutant seedlings, indicating that *rgs1-2* enhanced *bri1-301* and *bak1-4* phenotypes.

**BRI1 and BAK1 physically interact with G-protein subunits.** BRI1 and BAK1 function genetically with G-protein subunits to regulate sugar-responsive growth and development (Fig. 3). BRI1 and BAK1 are localized in the plasma membrane[28,35,36]. Similarly, G-protein subunits are associated with the plasma membrane to mediate sugar signaling[15]. In addition, it has been known that the defense-related receptor-like kinases such, as BAK1 and BIK1 interact with several G-protein subunits to influence defense responses[43,51]. We therefore asked whether BRI1 and BAK1 could directly interact with G-protein subunits to regulate sugar responses. As several studies showed that G-protein subunits can be expressed in *E. coli* although they function in a complex[12,43,52], we tested whether BRI1 and BAK1 could physically interact with G-protein subunits using in vitro pull-down assays. GPA1, AGB1, AGG1, AGG2, and AGG3 were expressed as maltose binding protein (MBP) fusion proteins in *E. coli*, while BRI1 kinase domain (BRI1-KD) and BAK1 kinase domain (BAK1-KD) were expressed as glutathione S-transferase (GST) fusion proteins in *E. coli*. GST-BRI1-KD bound to MBP-AGB1, MBP-AGG1, MBP-AGG2, and MBP-AGG3, but did not bind to MBP-GPA1 in our experimental conditions (Fig. 4a and

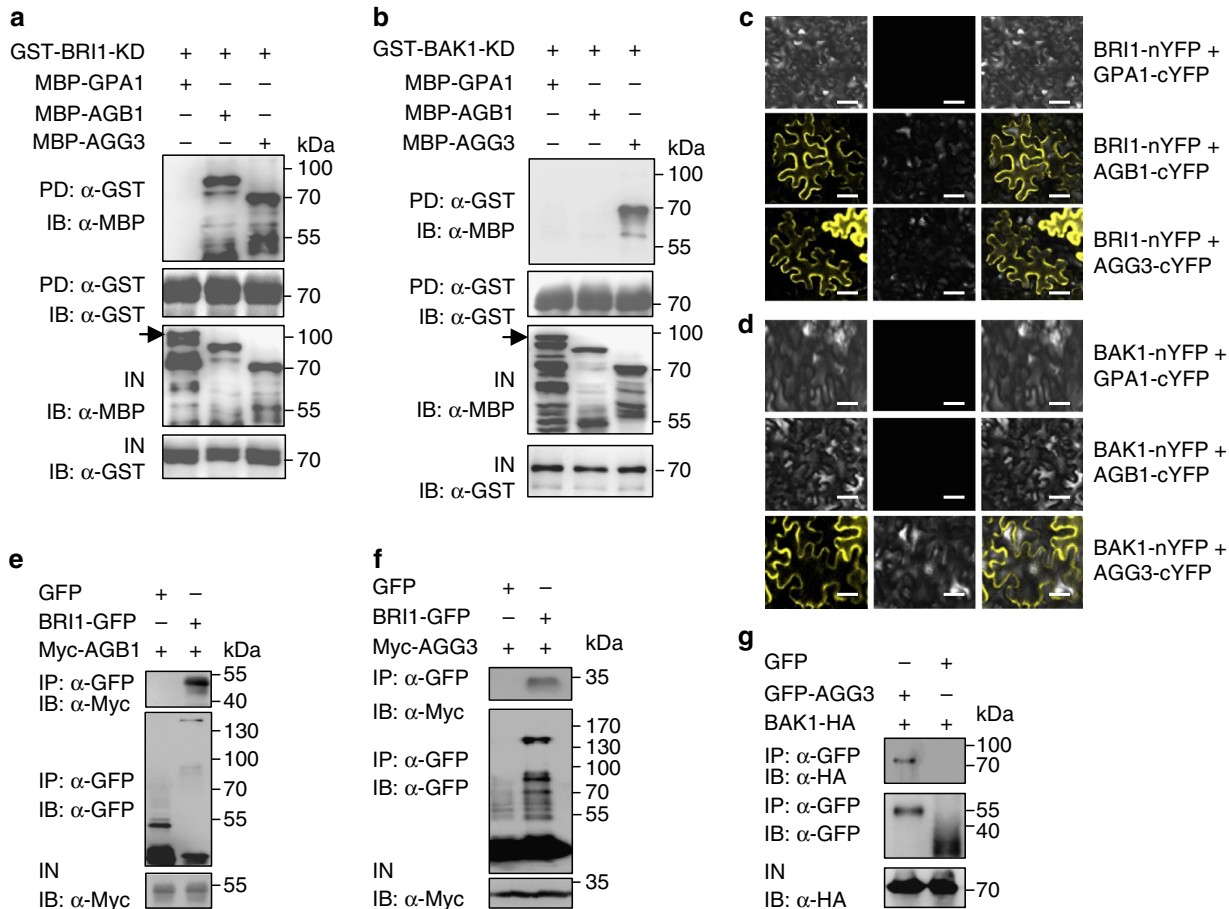

**Fig. 4** BRI1 and BAK1 physically interact with G proteins. **a** BRI1 kinase domain (BRI1-KD) interacts with AGB1 and AGG3 in vitro. MBP-GPA1, MBP-AGB1, or MBP-AGG3 was pulled down by GST-BRI1-KD and detected by immunoblot with anti-MBP antibody. Arrow indicates the band of MBP-GPA1.IN, input; IB, immunoblot. **b** BAK1 kinase domain (BAK1-KD) interacts with AGG3 in vitro. MBP-GPA1, MBP-AGB1, or MBP-AGG3 was pulled down by GST-BAK1-KD and detected by immunoblot with anti-MBP antibody. Arrow indicates the band of MBP-GPA1. IN, input; IB immunoblot. **c** Bimolecular fluorescence complementation (BiFC) assays show that BRI1 interacts with AGB1 and AGG3 in *N. benthamiana* leaves. BRI1-nYFP was coexpressed with GPA1-cYFP, AGB1-cYFP, or AGG3-cYFP in leaves of *N. benthamiana*. **d** BiFC assays show that BAK1 interacts with AGG3 in *N. benthamiana* leaves. BAK1-nYFP was coexpressed with GPA1-cYFP, AGB1-cYFP, or AGG3-cYFP in leaves of *N. benthamiana*. **e** BRI1 interacts with AGB1 in Arabidopsis. The extracted total proteins from *35S:GFP;35S:Myc-AGB1*, or *pBRI1:BRI1-GFP;35S:Myc-AGB1* seedlings were incubated with GFP-Trap-A, respectively. Immunoprecipitates were examined using anti-Myc and anti-GFP antibodies, respectively. IP immunoprecipitation; IN input; IB immunoblot. **f** BRI1 interacts with AGG3 in Arabidopsis. The extracted total proteins from *pBRI1:BRI1-GFP;35S:GFP* or *pBRI1:BRI1-GFP;35S:Myc-AGG3* seedlings were incubated with GFP-Trap-A, and examined using anti-Myc and anti-GFP antibodies, respectively. IP immunoprecipitation; IN input; IB immunoblot. **g** BAK1 interacts with AGG3 in Arabidopsis. The extracted total proteins from *35S:GFP;35S:BAK1-HA* and *35S:GFP-AGG3;35S:BAK1-HA* seedlings were incubated with GFP-Trap-A, and examined using anti-HA and anti-GFP antibodies, respectively. IP immunoprecipitation; IN input; IB immunoblot. Scale bars, 50 μm in **c**, **d**

Supplementary Fig.14a). By contrast, GST-BAK1-KD bound to MBP-AGG1, MBP-AGG2, and MBP-AGG3, but did not bind to MBP-GPA1 and MBP-AGB1 (Fig. 4b and Supplementary Fig. 14b). These results show that BRI1 and BAK1 directly interact with several G-protein subunits in vitro.

To further verify whether BRI1 and BAK1 physically associate with G-protein subunits in planta, we performed bimolecular fluorescence complementation (BiFC) assays. Coexpression of BRI1-nYFP with AGB1-cYFP, AGG1-cYFP, AGG2-cYFP, or AGG3-cYFP resulted in strong yellow fluorescent protein (YFP) fluorescence in leaves of *N. benthamiana* (Fig. 4c and Supplementary Fig. 14c). By contrast, we did not detect YFP fluorescence when BRI1-nYFP was coexpressed with GPA1-cYFP (Fig. 4c and Supplementary Fig. 14c). Similarly, YFP fluorescence was detected in coexpressions of BAK1-nYFP with AGG1-cYFP, AGG2-cYFP, or AGG3-cYFP, but not in GPA1-cYFP and AGB1-cYFP (Fig. 4d and Supplementary Fig. 14d), consistent with the results of in vitro pull-down assays.

We then performed co-immunoprecipitation (Co-IP) analyses to investigate the association of BRI1 with AGB1 and AGG3 in *Arabidopsis*. We generated *pBRI1:BRI1-GFP;35S:Myc-AGB1* and *pBRI1:BRI1-GFP;35S:Myc-AGG3* plants. Total proteins were isolated and incubated with green fluorescent protein (GFP)-Trap-A agarose beads to immunoprecipitate BRI1-GFP. Immunoprecipitated proteins were detected with anti-GFP and anti-Myc antibodies, respectively. As shown in Fig. 4e, f, Myc-AGB1 and Myc-AGG3 were detected in the immunoprecipitated BRI1-GFP complexes, indicating that BRI1 associates with AGB1 and AGG3 in *Arabidopsis*. Co-immunoprecipitation experiments also revealed that BAK1 associates with AGG3 in *Arabidopsis* (Fig. 4g).

**BRI1 and BAK1 phosphorylate G-protein subunits**. As BRI1 has kinase activity and interacts with AGB1, AGG1, AGG2, and AGG3, we asked whether BRI1 is required for the

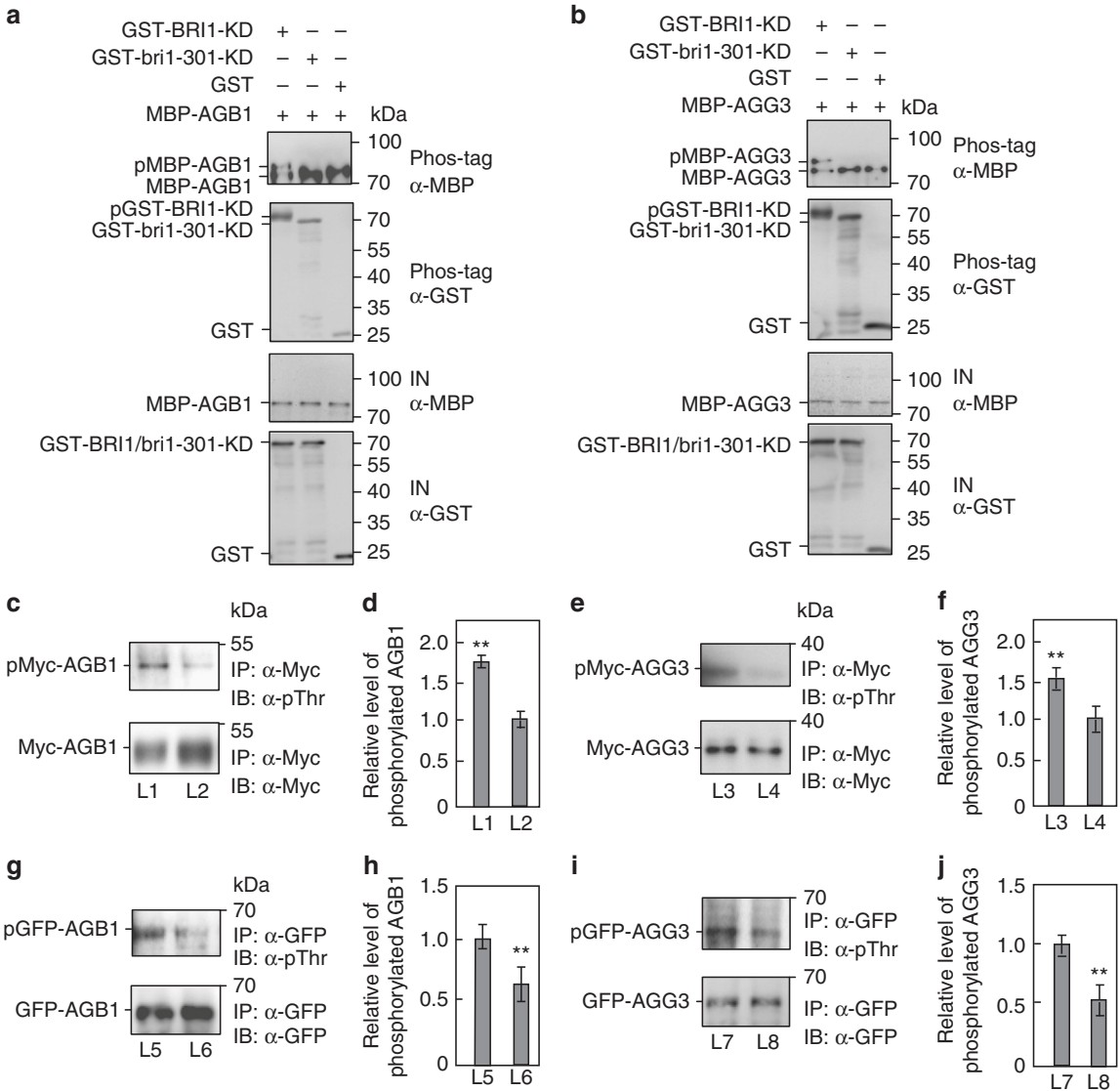

**Fig. 5** BRI1 is required for the phosphorylation of AGB1 and AGG3. **a** BRI1 kinase domain (BRI1-KD) phosphorylates AGB1 in vitro, but bri1-301 kinase domain (bri1-301-KD) does not. The phosphorylated MBP-AGB1 (pMBP-AGB1), unphosphorylated (MBP-AGB1) and autophosphorylated GST-BRI1-KD (pGST-BRI1-KD) were separated by phos-tag SDS-PAGE. IN, input. **b** BRI1-KD phosphorylates AGG3 in vitro, but bri1-301-KD does not. The phosphorylated MBP-AGG3 (pMBP-AGG3), unphosphorylated (MBP-AGG3) and autophosphorylated GST-BRI1-KD (pGST-BRI1-KD) were separated by phos-tag SDS-PAGE. IN, input. **c, d** BRI1 is required for the phosphorylation of AGB1 in *Arabidopsis*. Total proteins from *pBRI1:BRI1-GFP;35S:Myc-AGB1* (L1) and *35S:GFP;35S:Myc-AGB1* (L2) seedlings were mixed with anti-Myc-Tag mouse mAb conjugated agarose beads, and detected using anti-Myc and anti-phosphothreonine (pThr) antibodies, respectively **c**. Quantification of the phosphorylated Myc-AGB1 was relative to the immunoprecipitated Myc-AGB1 **d**. Values are given as mean ± SD ($n = 3$). **$P < 0.01$ compared with the value for L2 seedlings by Student's $t$-test. **e, f** BRI1 is required for the phosphorylation of AGG3 in *Arabidopsis*. Total proteins from *pBRI1:BRI1-GFP;35S:Myc-AGG3* (L3) and *35S:GFP;35S:Myc-AGG3* (L4) seedlings were mixed with anti-Myc-Tag mouse mAb conjugated agarose beads and detected using anti-Myc and anti-phosphothreonine (pThr) antibodies, respectively (**e**). Quantification of the phosphorylated Myc-AGG3 was relative to the immunoprecipitated Myc-AGG3 **f**. Values are given as mean ± SD ($n = 3$). **$P < 0.01$ compared with the value for L4 seedlings by Student's $t$-test. **g, h** BRI1 is required for the phosphorylation of AGB1 in *Arabidopsis*. Total proteins from *35S:GFP-AGB1* (L5) and *35S:GFP-AGB1;bri1-301* (L6) seedlings were incubated with GFP-Trap-A, and detected using anti-GFP and anti-phosphothreonine (pThr) antibodies, respectively **g**. Quantification of the phosphorylated GFP-AGB1 was relative to the immunoprecipitated GFP-AGB1 **h**. Values are given as mean ± SD, ($n = 3$). **$P < 0.01$ compared with the value for L5 seedlings by Student's $t$-test. **i, j** BRI1 is required for the phosphorylation of AGG3 in *Arabidopsis*. Total proteins from *35S:GFP-AGG3* (L7) and *35S:GFP-AGG3;bri1-301* (L8) seedlings were incubated with GFP beads, and detected by immunoblot with anti-GFP and anti-phosphothreonine (pThr) antibodies, respectively **i**. Quantification of the phosphorylated GFP-AGG3 was relative to the immunoprecipitated GFP-AGG3 **j**. Values are given as mean ± SD ($n = 3$). **$P < 0.01$ compared with the value for L7 seedlings by Student's $t$-test

phosphorylation of AGB1, AGG1, AGG2, and AGG3. GST-fused BRI1 kinase domain (GST-BRI1-KD) and GST-fused bri1-301 kinase domain (GST-bri1-301-KD) were incubated with MBP-AGB1, MBP-AGG1, MBP-AGG2, or MBP-AGG3 in an in vitro kinase assay buffer, respectively. The autophosphorylation activity was detected in the presence of GST-BRI1-KD but not GST-

bri1-301-KD (Fig. 5a, b and Supplementary Fig. 15a,b), consistent with a previous study[53]. The phosphorylated MBP-AGB1, MBP-AGG1, MBP-AGG2, and MBP-AGG3 were detected in the presence of GST-BRI1-KD, while the phosphorylated MBP-AGB1, MBP-AGG1, MBP-AGG2, and MBP-AGG3 were not observed in the presence of GST-bri1-301-KD or GST (Fig. 5a, b and

**a**
1-MSVSELKERHAVA**T(14)**ET**(16)**VNNLRDQLRQRRLQLLD**T(34)**DVARY**S(40)**AAQ

GR**T(46)**RV**S(49)**FGA**T(53)**DLVCCRTLQGH**T(65)**GKVY**S(70)** LDWTPERNRIV**S(82)**

ASQDGRLIVWNALTSQK**T(100)**HAIKL…SRAVR**T(243)**FHGHEGDVN**T(253)**VKFFP

DGYRFG**T(265)**GSDD…GGHRRVI-377

**b**
1-MSAPSGGGEGGGKESAAGGV**S(21)S(22)**SSLAPSSLPPPRPK**S(37)**PPEYP…EG

EIKFIEGVQPA**S(78)**RC IKEVSDFVVA N**S(92)**DPLIPAQRKSRRS…CNC**T(143)**SCS

CIGSK…CFRSCSC**T(199)**RPS…SNPCCLAF-251

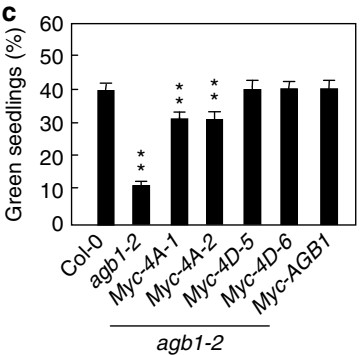

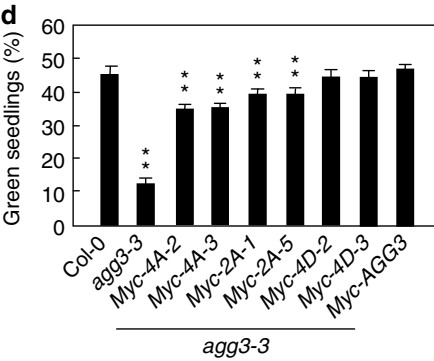

**Fig. 6** Phosphorylations of AGB1 and AGG3 affect sugar responses. **a**, **b** Detection of phosphorylation sites of AGB1 (**a**) and AGG3 (**b**) by LC–MS/MS after in vitro phosphorylation reaction. AGB1 and AGG3 contain 377 and 251 residues, respectively (The omitted residues of AGB1 and AGG3 were indicated by ···). The phosphorylated residues detected by LC–MS/MS were shown in red. The possible phosphorylated residues, but not detected by LC–MS/MS, were shown in blue. Four primary phosphorylatable residues shown by underline, were substituted into phospho-dead or phospho-mimicking residues, respectively. **c** The percentage of the indicated seedlings with green cotyledons. *agb1-2* mutants were transformed with *35S:Myc-AGB1*, *35S:Myc-AGB1*[T14A,S40A,T243A,T253A] (*Myc-4A-1* and *Myc-4A-2*), or *35S:Myc-AGB1*[T14D,S40D,T243D,T253D] (*Myc-4D-5* and *Myc-4D-6*). Independent T2 transgenic lines were selected for assay. Seedlings were grown on medium supplemented with 6% glucose medium under constant light condition for 10 days ($n \geq 69$). **d** The percentage of the indicated seedlings with green cotyledons. *agg3-3* mutants were transformed with *35S:Myc-AGG3*, *35S:Myc-AGG3*[S21A,S22A] (*Myc-2A-1*, and *Myc-2A-5*), *35S:Myc-AGG3*[S21A,S22A,T143A,T199A] (*Myc-4A-2* and *Myc-4A-3*), or *35S:Myc-AGG3*[S21D,S22D,T143D,T199D] (*Myc-4D-2* and *Myc-4D-3*). Independent T2 transgenic lines were selected for assay. Seedlings were grown on 6% glucose medium under constant light condition for 10 days ($n \geq 74$). Values in **c**, **d** are given as mean ± SD, \*\*$P < 0.01$ compared with Col-0 by Student's *t*-test

Supplementary Fig. 15a,b). These results show that BRI1 can phosphorylate AGB1, AGG1, AGG2, and AGG3 in vitro. Similarly, GST-fused BAK1 kinase domain (GST-BAK1-KD) can phosphorylate MBP-AGG1, MBP-AGG2, or MBP-AGG3 in vitro (Supplementary Fig. 16a-c).

To verify that BRI1 is required for the phosphorylation of AGB1 and AGG3, we examined the phosphorylation levels of AGB1 and AGG3 in *pBRI1:BRI1-GFP;35S:Myc-AGB1*, *35S: GFP;35S:Myc-AGB1*, *pBRI1:BRI1-GFP;35S:Myc-AGG3* and *35S: GFP;35S:Myc-AGG3* plants, respectively. *pBRI1:BRI1-GFP* transgenic plants showed long petioles and dome-shaped leaves (Supplementary Fig. 17), like those observed in plants over-expressing *BRI1*[54], indicating that *pBRI1:BRI1-GFP* transgenic plants had the increased BRI1 activity. As shown in Fig. 5c, d and Supplementary Fig. 18a, *pBRI1:BRI1-GFP;35S:Myc-AGB1* seedlings had more phosphorylated Myc-AGB1 than *35S:GFP;35S: Myc-AGB1* seedlings. Similarly, *pBRI1:BRI1-GFP;35S:Myc-AGG3* plants contained more phosphorylated Myc-AGG3 than *35S: GFP;35S:Myc-AGG3* plants (Fig. 5e, f and Supplementary Fig. 18b). These data indicate that the increased BRI1 activity promotes the phosphorylation of AGB1 and AGG3 in *Arabidopsis*. We further investigated the phosphorylation levels of AGB1 and AGG3 in *bri1-301*. As shown in Fig. 5g–j and Supplementary Fig. 18c,d, the phosphorylation levels of AGB1 and AGG3 were reduced in *bri1-301* compared with those in the wild type. These results reveal that BRI1 is required for the phosphorylation of AGB1 and AGG3 in *Arabidopsis*.

**Phosphorylations of AGB1 and AGG3 affect sugar responses**. Considering that BRI1 can phosphorylate AGB1 and AGG3 in vitro and in vivo, we investigated phosphorylation sites of AGB1 and AGG3 by LC–MS/MS analysis. As shown in Fig. 6a and Supplementary Table 1, we identified 25 phosphopeptides of AGB1, which correspond to 14 phosphosites. Interestingly, we observed that the peptides with amino acids T14, S40, T243, or T253 accounted for 60% of the phosphopeptides identified by LC–MS/MS analysis (Supplementary Table 1). We further investigated the role of these four major phosphorylation sites in regulating sugar responses. These four amino acids were simultaneously substituted into phospho-dead alanine (AGB1[T14A,S40A, T243A,T253A]) or phospho-mimicking aspartic acid (AGB1[T14D, S40D,T243D,T253D]) (Fig. 6a). We then introduced these phospho-dead and phospho-mimicking forms of AGB1 into *agb1-2* plants. As shown in Fig. 6c, the green cotyledon percentages of transgenic lines overexpressing AGB1 or phospho-mimicking forms of AGB1 in *agb1-2* mutant were similar to those of the wild type when they were grown on medium with 6% glucose, while the green cotyledon percentages of transgenic plants overexpressing AGB1[T14A,S40A,T243A,T253A] in *agb1-2* mutant were significantly lower than those of the wild type. These results indicate that the phosphorylations of amino acids T14, S40, T243, and T253 in AGB1 are important for sugar responses in *Arabidopsis*.

We then identified phosphorylation sites in AGG3 by LC–MS/MS analysis (Supplementary Table 2). We found 5 phosphopeptides and the peptides with amino acids S21 or S22 accounted for 50% of the phosphopeptides (Fig. 6b and Supplementary Table 2).

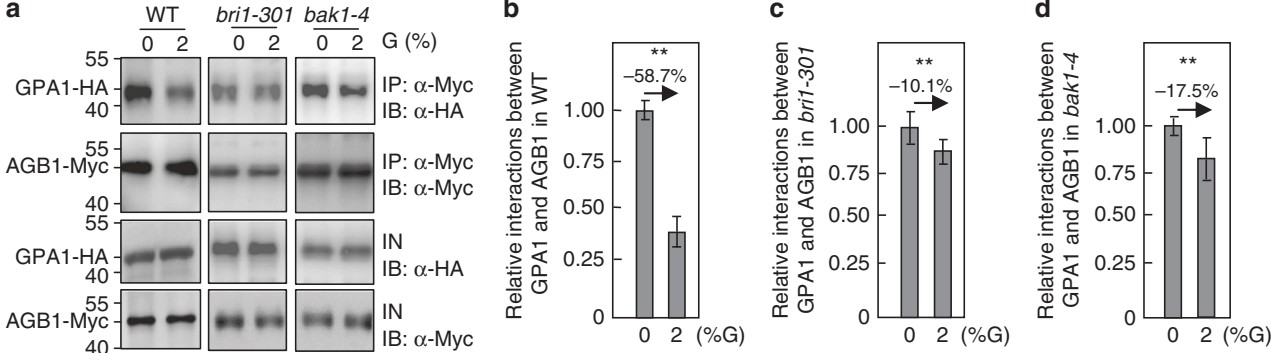

**Fig. 7** BRI1 and BAK1 affect the interactions between GPA1 and AGB1. **a** BRI1 and BAK1 influences the interactions between GPA1 and AGB1 in *Arabidopsis*, respectively. The wild type, *bri1-301*, and *bak1-4* leaf protoplasts co-transformed with *35S:AGB1-Myc* and *35S:GPA1-HA* plasmids were grown in darkness for 14 h, and then treated without or with 2% glucose (G) for 5 h. Total proteins from these leaf protoplasts were incubated with anti-Myc-Tag mouse mAb conjugated agarose beads, and detected using anti-Myc and anti-HA antibodies, respectively. IN input; IP immunoprecipitation. **b–d** Quantification of relative interactions between GPA1 and AGB1 in wild type (**b**), *bri1-301* (**c**) and *bak1-4* (**d**) leaf protoplasts treated without or with 2% glucose as shown in **a**. The ratio was shown as immunoprecipitated GPA1-HA to input GPA1-HA. Values are given as mean ± SD ($n = 3$) relative to the value for 0% glucose, set at 1. Values in **b–d** are given as mean ± SD, **$P < 0.01$ compared with the value for 0% glucose treatment by Student's *t*-test

Unfortunately, we could not identify phosphopeptides with threonine although AGG3 contains two threonines (T143 and T199).

It is possible that the phosphopeptides with threonine had lower abundance in our in vitro phosphorylation conditions. The increased BRI1 activity promotes the phosphothreonine level of AGG3, while the decreased BRI1 activity inhibits the phospho-threonine level of AGG3 in *Arabidopsis* (Fig. 5e, f, i, j), indicating that T143 and/or T199 of AGG3 could be phosphorylated in vivo. We therefore mutated amino acids S21, S22, T143, and T199 into phospho-dead alanine (AGG3^{S21A,S22A}; AGG3^{S21A,S22A,T143A,T199A}) or phospho-mimicking aspartic acid (AGG3^{S21D,S22D,T143D,T199D}). We then introduced these phospho-dead and phospho-mimicking forms of AGG3 into *agg3-3* plants. Phenotype analyses suggest that the phosphorylations of amino acids S21, S22, T143, and T199 in AGG3 are important for sugar responses in *Arabidopsis* (Fig. 6d).

**BRI1 and BAK1 affect the interactions between GPA1 and AGB1.** Previous studies have proposed that glucose promotes the heterotrimer dissociation of G proteins and causes free Gα and Gβ/γ subunits, which interact with the downstream effectors, thereby transmitting glucose signals to multiple intracellular signaling[15,55]. As BRI1 interacts with and phosphorylates G-protein subunits, we asked whether BRI1 could affect the interaction between GPA1 and AGB1, which is crucial for the activation of G-protein signaling. To test this, we transiently coexpressed *35S:GPA1-HA* and *35S:AGB1-Myc* in *Arabidopsis* leaf protoplasts treated without or with glucose. As shown in Fig. 7a–d, the interactions between GPA1 and AGB1 in wild-type seedlings were dramatically reduced in response to exogenous glucose. By contrast, their interactions in *bri1-301* and *bak1-4* seedlings were not strongly decreased in response to glucose. These results suggest that BRI1 and BAK1 may influence the dissociation of Gα and Gβ/γ subunits in response to glucose.

## Discussion

Sugar sensing and signaling are crucial for plant growth and development, gene expression, and metabolic processes[1,4,56–58]. Crosstalks between sugar and hormone signaling pathways play essential roles in balancing carbon availability and plant growth and development. In this study, we show that the brassinosteroid receptors play key roles in regulating sugar-responsive growth

and development in *Arabidopsis*. Biochemical results show that BRI1 and BAK1 physically interact with G proteins. BRI1 can phosphorylate AGB1 and AGGs, and BAK1 can also phosphorylate AGGs. Genetic analyses suggest that BRI1 and BAK1 function in a common pathway with G proteins to regulate sugar responses. Thus, our findings define an important genetic and molecular mechanism by which BR receptors interact with G proteins to regulate sugar-responsive growth and development in *Arabidopsis*.

The dark development is a sensitive indicator of the effects of sugars on growth and development in *Arabidopsis*[32–34]. In this study, we found that dark-grown *bri1-301* and *bak1-4* mutants exhibited delayed development in response to exogenous glucose (Fig. 1b), indicating that BRI1 and BAK1 are important for sugar-responsive growth and development in the dark. Consistent with this, dark-grown *35S:BRI1* showed the slightly enhanced development in response to glucose (Supplementary Fig.19). By contrast, dark-grown *det2* mutant exhibited the delayed development in response to exogenous glucose, suggesting that a proper level of BR is also important for sugar-responsive growth and development in the dark (Supplementary Fig.19). Hypocotyl elongation of dark-grown *Arabidopsis* seedlings was promoted by low concentrations of glucose, while hypocotyl elongation was progressively inhibited by high concentrations of glucose[33,34]. Dark-grown *bri1-301* and *bak1-4* mutants showed glucose-insensitive hypocotyl elongation (Supplementary Fig. 2). In light, high concentrations of glucose-induced developmental arrest during early seedling morphogenesis[4,20]. Light-grown *bri1-301* and *bak1-4* mutants exhibited insensitivity to high-glucose-induced developmental arrest (Fig. 1c). These results demonstrate that BRI1 and BAK1 play key roles in regulating sugar-responsive growth and development under both light and dark conditions.

BRI1 usually associates with its co-receptor BAK1 to activate BR signaling[30,35,36]. Here we found that glucose influences the physical associations between BRI1 and BAK1 in a glucose concentration-dependent manner. The interactions between BRI1 and BAK1 were progressively increased in response to low glucose, and then gradually decreased in response to high glucose (Fig. 2a). It has been known that low concentrations of glucose promote *Arabidopsis* seedling growth, and high concentrations of glucose repress *Arabidopsis* seedling growth. Thus, these findings suggest that the physical interactions between BRI1 and BAK1 are important for sugar-responsive growth and development. The leucine-rich repeat receptor kinases flagellin-sensitive 2 (FLS2)

has been reported to interact with BAK1 to regulate defence responses[59,60]. However, glucose did not influence the interactions between FLS2 and BAK1 (Supplementary Fig.20), suggesting that the interactions of FLS2 and BAK1 might not be involved in sugar responsive growth and development. We also observed that the BRI1 and BAK1 interactions induced by low glucose were obviously decreased when BRZ was added for 6 h (Supplementary Fig.21a). The BRI1 and BAK1 interactions were almost abolished when we extended the BRZ treatment time (Supplementary Fig.21b). These results suggest that a proper level of BR is required for the BRI1 and BAK1 interactions induced by low glucose in plants. The interactions of BRI1 with BAK1 usually result in their phosphorylation, which is crucial for phosphorylating downstream components[35,37]. Interestingly, we observed that glucose influences the phosphorylation levels of BRI1 and BAK1 in a concentration-dependent manner (Fig. 2c–f). In addition, low glucose can induce interactions and phosphorylations of BRI1 and BAK1 in a short time (Supplementary Figure 22). Thus, glucose regulates the interaction and phosphorylation of BRI1 and BAK1, which might be important to mediate sugar responses.

We found that the G-protein pathway is required for BR and sugar crosstalks in regulating dark development and high-glucose-induced developmental arrest (Supplementary Figure 9 and 11). However, hypocotyls of G-protein mutants showed roughly similar response to those of the wild type (Supplementary Figure 10)[24]. Previous studies showed that BR and sugar crosstalks act through the HXK1 pathway to regulate hypocotyl elongation in plants[24]. These results suggest that BR and sugar might act through G proteins to regulate different processes of seedling development through different mechanisms.

We further found that BRI1 physically interacts with AGB1 and AGGs, and BAK1 associates with AGGs in vitro and in vivo. Interestingly, a previous study indicated that BAK1 interacts with GPA1 but not AGB1 and AGG3 in yeast cells, and these interactions have been proposed to mediate defense responses[51]. In this study, we firstly showed that BAK1 physically interacts with AGG3 using in vitro pull-down assays (Fig. 4b). We then indicated that BAK1 is associated with AGG3 using bimolecular fluorescence complementation (BiFC) assays (Fig. 4d). Finally, we revealed that BAK1 is associated with AGG3 in Arabidopsis using co-immunoprecipitation (Co-IP) analyses (Fig. 4g). Therefore, these results strongly demonstrated that BAK1 physically associates with AGG3 in vitro and planta. By contrast, we did not detect interactions between BAK1 and GPA1 using in vitro pull-down assays and in vivo BiFC assays (Fig. 4b, d); although, we carefully repeated these experiments for several times. One possible explanation is that the interactions between BAK1 and GPA1 might be too weak to be detected in our experimental conditions or their interactions might depend on specific conditions. As BRI1 and BAK1 interact with several G-protein subunits (Fig. 4), BRI1 and BAK1 may regulate sugar responses, at least in part, by influencing G-protein signaling. Consistent with this, we observed that the interactions between BRI1 and AGB1 were increased in response to 2% glucose (Supplementary Fig. 23). It is possible that the interactions of BRI1 with G-protein subunits could affect the phosphorylation of G-protein subunits. Supporting this idea, we detected that BRI1 phosphorylates AGB1 and AGGs in vitro. Overexpression of BRI1 increased the phosphorylation levels of AGB1 and AGG3 in Arabidopsis, while the phosphorylation levels of AGB1 and AGG3 were reduced in bri1-301 mutant (Fig. 5 and Supplementary Figure 18). Similarly, a recent study showed that the FLS2-BIK1 immune receptor complex interacts with and phosphorylates an extra-large isoform of the Gα (XLG2), thereby modulating immune responses[43]. It is plausible that G proteins might interact with different plasma membrane receptor-like kinases to mediate a variety of plant growth and development processes and stress responses in plants. Previous studies suggested that glucose promotes the heterotrimer dissociation of G proteins and causes free Gα and Gβ/γ subunits, resulting in the activation of G-protein signaling[13,15,55]. Recently, BIK1 has been reported to phosphorylate an extra-large isoform of the Gα, resulting in the dissociation of Gα and Gβ/γ and the activation of G-protein signaling[43]. We observed that the bri1-301 and bak1-4 mutations influence the interactions between GPA1 and AGB1 in response to exogenous glucose (Fig. 7). Consistent with this, BRZ promotes the interactions between Gα and Gβ/γ (Supplementary Figure 24). Thus, it is possible that the phosphorylation of AGB1 and AGGs by BRI1 and BAK1 may cause the dissociation of Gα and Gβ/γ, which activates G-protein signaling. In consistent with biochemical data, our genetic analyses showed that gpa1-101, agb1-2, and agg3-3 suppressed the dark development and high-glucose-induced growth arrest phenotypes of bri1-301 and bak1-4 (Fig. 3). By contrast, bri1-301 and bak1-4 enhanced the glucose-insensitive phenotype of rgs1-2 (Supplementary Fig. 12 and 13), suggesting that BR receptors might act together[61] or redundantly with RGS1 to regulate G-protein-mediated sugar responses. Interestingly, BRI1 and BAK1 can interact with RGS1 in vitro and in vivo, respectively (Supplementary Fig. 25a-c), consistent with a previous result[50]. Together, sugars influence physical interactions and phosphorylations of BRI1 and BAK1 in a concentration-dependent manner. The activated BRI1/BAK1 complex can phosphorylate and activate G-protein signaling, which finally regulates sugar-responsive growth and development (Supplementary Figure 26).

## Methods

**Plant materials and growth conditions.** The bri1-301, bak1-4, gpa1-101, agb1-2, gpa1-101, agg3-3, rgs1-2, pBRI1;BRI1-GFP, and 35S:BAK1-HA have been used in previous studies[28,46,53,62–65]. Arabidopsis seeds were sterilized with 10% bleach for 10 min, washed with water three times, and then stored at 4 °C for 3 days. For dark development assay, seeds were dispersed on MS medium without or with 1, 2, 4, and 6% glucose, irradiated under constant light for 18 h, and then grown vertically in the dark condition at 22 °C. For green seedling establishment assay, seeds were grown on MS medium with 6% glucose under constant light condition at 22 °C. Seedlings with green cotyledon were used to count statistics.

**In vitro pull-down assay.** The coding sequences of BRI1 kinase domain (residues 814–1196) and BAK1 kinase domain (residues 250–662) were cloned into the pGEX-4T-1 plasmid to make GST-BRI1-KD and GST-BAK1-KD constructs, respectively. Primers used for these two constructs were GST-BRI1-KD-FP/RP and GST-BAK1-KD-FP/RP, respectively (Supplemental Table 3). Similarly, GPA1, AGB1, AGG1, AGG2, AGG3, and RGS1 were cloned into the EcoR I and Pst I sites of the pMAL-c2 plasmid to create MBP-GPA1, MBP-AGB1, MBP-AGG1, MBP-AGG2, MBP-AGG3, and MBP-RGS1 constructs, respectively. Primers used for MBP-GPA1, MBP-AGB1, MBP-AGG1, MBP-AGG2, MBP-AGG3, and MBP-RGS1 constructs were MBP-GPA1- FP/RP, MBP-AGB1- FP/RP, MBP-AGG1-FP/RP, MBP-AGG2- FP/RP, MBP-AGG3-FP/RP, and MBP-RGS1, respectively (Supplemental Table 3).

To test whether BRI1 and BAK1 interact with G proteins, about 15 µg GST-BRI1-KD or GST-BAK1-KD proteins were incubated with about 30 µg MBP-GPA1, MBP-AGB1, MBP-AGG1, MBP-AGG2, MBP-AGG3, or MBP-RGS1 proteins in 900 µL TGH buffer (10% glycerol, 1% Triton X-100, 1.5 mM MgCl$_2$, 150 mM NaCl, 1 mM EGTA, 50 mM HEPES, pH 7.5, 1 mM PMSF and 1× complete protease inhibitor cocktail). Similarly, about 15 µg proteins were incubated with about 30 µg MBP-GPA1, MBP-AGB1, MBP-AGG1, MBP-AGG2, MBP-AGG3, or MBP-RGS1 fusion proteins, respectively. A volume of 30 µL GST beads were incubated with each reaction mixture with slowly shaking for about 1 h. After reaction, beads were washed three times and heated for 10 min in a 98 °C metal bath. The immunoprecipitated proteins were separated by SDS-PAGE electrophoresis and detected by anti-GST (Abmart M20007, 1/5000) and anti-MBP (New England Biolabs E8032, 1/10,000) antibodies, respectively. Pull-down-related blots were shown in Supplementary Figrue28, 32 and 35.

**In vivo co-immunoprecipitation.** The coding sequences (CDSs) of AGB1, AGG3, and RGS1 were amplified and cloned into pCAMBIA1300-221-Myc plasmid to create 35S:Myc-AGB1, 35S:Myc-AGG3, and 35S:Myc-RGS1 constructs, respectively. Primers used for these three constructs were listed in Supplemental Table 3. Total proteins from pBRI1:BRI1-GFP;35S:Myc-AGB1, 35S:GFP;35S:Myc-AGB1, pBRI1:

*BRI1-GFP;35S:Myc-AGG3, 35S:GFP;35S:Myc-AGG3, pBRI1:BRI1-GFP;35S:Myc-RGS1, 35S:GFP;35S:Myc-RGS1,* and *pBAK1:BAK1-GFP;35S:Myc-RGS1* seedlings were isolated with the extraction buffer (20% glycerol, 2% Triton X-100, 1 mM EDTA,150 mM NaCl, 50 mM Tris-HCl, pH 7.5, 1 mM PMSF, and 1× complete protease inhibitor cocktail). GFP-Trap-A agarose were used in the subsequent Co-IP assays. The immunoprecipitates were detected by anti-Myc (Abmart M20002, 1/5000) and anti-GFP (Abmart M20004, 1/5000) antibodies, respectively. Similarly, *35S:GFP;35S:BAK1-HA* and *35S:GFP-AGG3;35S:BAK1-HA* seedlings were used for Co-IP assays and immunoblots were performed using anti-HA (Abmart M20003, 1/5000) and anti-GFP (Abmart M20004, 1/5000) antibodies, respectively.

To examine the interactions between BRI1 and BAK1 using their native antibodies, wild-type seedlings were isolated with the extraction buffer, and incubated with anti-BAK1 antibody for 30 min, and then incubated with Protein G Magnetic Beads for 30 min to immunoprecipitate BAK1. Immunoprecipitated BAK1 proteins were examined using anti-BAK1 (Abiocode R2350-3, 1/500) and anti-BRI1 (Abiocode R3283-4, 1/500) antibodies, respectively.

Wild type, *bak1-4*, and *bri1-301* were grown in the light condition for 17 days, and then incubated in darkness for 3 days. After the treatment, the leaves of these seedlings were collected to generate protoplasts. *Arabidopsis* leaf protoplasts of the wild type, *bak1-4* and *bri1-301* were co-transformed with *35S:AGB1-Myc* and *35S:GPA1-HA* plasmids, then grown in darkness for 14 h, and treated without or with 2% glucose for 5 h. Total proteins from wild type and *bri1-301* leaf protoplasts treated without or with 2% glucose were isolated with the extraction buffer. Anti-Myc-Tag mouse mAb conjugated agarose beads were used in the subsequent Co-IP experiments, and the immunoprecipitates were examined using anti-Myc (Abmart M20002, 1/5000) and anti-HA (Abmart M20003, 1/5000) antibodies, respectively. Co-immunoprecipitation-related blots were shown in Supplementary Figrue 27, 28, 30, 31, 33, 34 and 35.

**Confocal microscopy and fluorescence quantification.** Fluorescence was observed using a Zeiss LSM710 META laser scanning microscope. Wavelengths of 488 nm and 520 nm were used for the observations of GFP and YFP signals, respectively. Fluorescence images were captured by the ZEN2009 software and fluorescence quantity was acquired by the ImageJ software[15]. At least 40 seedlings were used for fluorescence quantity measurements of each value. Three independent biological experiments were repeated for these fluorescence quantifications.

**Bimolecular fluorescence complementation analysis.** To create *BRI1-nYFP* and *BAK1-nYFP* constructs, CDSs of *BRI1* and *BAK1* were cloned into the *35S:pUC-SPYNE* vector, respectively. Similarly, CDSs of *GPA1, AGB1, AGG1, AGG2, AGG3,* and *RGS1* were cloned into the *35S:pUC-SPYCE* vector to generate *GPA1-cYFP, AGB1-cYFP, AGG1-cYFP, AGG2-cYFP, AGG3-cYFP,* and *RGS1-cYFP* constructs, respectively. The specific primers used for *BRI1-nYFP, BAK1-nYFP, GPA1-cYFP, AGB1-cYFP AGG3-cYFP,* and *RGS1-cYFP* constructs were described in Supplemental Table 3. These constructs were subsequently introduced into *Agrobacterium* strain GV3101. *Agrobacterium* strains containing different nYFP and cYFP pairs were co-transformed into leaves of *N. benthamiana*. After transformation, plants were grown in light condition for 72 h before observation. The YFP fluorescence was observed using a Zeiss LSM710 microscope (Zeiss LSM 710).

**Phosphorylation assays.** The coding sequence of bri1-301 kinase domain (residues 814–1196) was cloned into the *pGEX-4T-1* plasmid to create GST-bri-301-KD construct. Primers for GST-bri-301-KD construct were GST-bri-301-KD-FP/RP (Supplementary Table 3). For in vitro phosphorylation assays, 200 ng GST, GST-BRI1-KD or GST-bri-301-KD proteins were incubated with 2 mg MBP-AGB1, MBP-AGG1, MBP-AGG2, MBP-AGG3, or MBP-RGS1 in 20 μl kinase reaction buffer (10 mM MgCl₂, 25 mM Tris-HCl, pH 7.5, 100 mM ATP, and 1 mM DTT) at 30 °C for 3 h. Similarly, GST-BAK1-KD proteins were incubated with MBP-AGG1, MBP-AGG2, MBP-AGG3, or MBP-RGS1 in kinase reaction buffer at 30 °C for 3 h. After that, SDS loading buffer was used to stop the kinase reactions. To separate the phosphorylated proteins, 80 μM Phos-tag (Wako) and 160 μM MnCl₂ were added to 10% SDS-PAGE gel, and the electrophoreses were performed at 40 volts for 12 h. The gels were washed with 10 mM EDTA (pH 8.0) for five times to remove the Mn²⁺, and then washed with transfer buffer (0.027 M Tris, 0.19 M glycine, and 3.47 M ethanol) for three times before wet electroblotting. The phosphorylated MBP-AGG1, MBP-AGG2, MBP-AGG3, or MBP-RGS1 proteins were examined using anti-MBP antibody (New England Biolabs E8032, 1/10,000).

Extracted proteins from *pBRI1;BRI1-GFP* and *pBAK1;BAK1-GFP* seedings were mixed with GFP-Trap-A (Chromotek) for 1 h at 4 °C, respectively. The immunoprecipitates were seperated by SDS-PAGE and detected by anti-phosphoserine (Sigma P3430, 1/200), or anti-phosphothreonine (Cell Signaling 9381, 1/500) and anti-GFP (Abmart M20004, 1/5000) antibodies, respectively.

To examine the phosphorylation levels of BRI1 and BAK1 using their native antibodies, wild-type seedings were isolated with the extraction buffer, and incubated with anti-BRI1 (Abiocode R3283-4, 1/500) or anti-BAK1 (Abiocode R2350-3, 1/500) antibodies for 30 min, then incubated with Protein G Magnetic Beads for 30 min to immunoprecipitate BRI1 and BAK1, respectively. Immunoprecipitated BRI1 proteins were examined by anti-BRI1 (Abiocode R3283-4, 1/500) and anti-phosphoserine (Sigma P3430, 1/200) antibodies, respectively.

Immunoprecipitated anti-BAK1 proteins were detected using anti-BAK1 (Abiocode R2350-3, 1/500) and anti-phosphoserine (Sigma P3430, 1/200) antibodies, respectively.

To examine the phosphorylation levels of AGB1 in *pBRI1:BRI1-GFP;35S:Myc-AGB1* and *35S:GFP;35S:Myc-AGB1* seedlings, extracted proteins from these seedlings were mixed with anti-Myc-Tag mouse mAb conjugated agarose beads (Abmart M20013) for about 1 h at 4 °C. The immunoprecipitates were seperated using electrophoresis and detected by anti-phosphothreonine (anti-pThr) (Cell Signaling, 9381, 1/500) or anti-phosphoserine (anti-pSer) (Sigma P3430, 1/200), and anti-Myc (Abmart M20002, 1/5000) antibodies, respectively. Similarly, phosphorylation levels of AGG3 were examined in *pBRI1:BRI1-GFP;35S:Myc-AGG3* and *35S:GFP;35S:Myc-AGG3* seedlings,

To examine the phosphorylation levels of AGB1 in *35S:GFP-AGB1, 35S:Myc-AGB1, 35S:GFP-AGB1;bri1-301,* and *35S:Myc-AGB1;bri1-301* seedlings, total proteins from these seedlings were isolated with the extraction buffer, and mixed with GFP-Trap-A for 1 h at 4 °C. The immunoprecipitates were seperated using electrophoresis and detected by anti-phosphothreonine (anti-pThr) (Cell Signaling 9381, 1/500) and anti-GFP (Abmart M20004, 1/5000) antibodies, respectively. Similarly, phosphorylation levels of AGG3 were examined in in *35S:GFP-AGG3, 35S:Myc-AGG3, 35S:GFP-AGG3;bri1-301,* and *35S:Myc-AGG3;bri1-301* seedlings. Phosphorylation-related blots were shown in Supplementary Figrue 27, 29, 31, 32, 33 and 34.

**Quantification of protein levels.** To study the effects of glucose on the interactions between BRI1 and BAK1, The 5-day-old seedlings of *pBRI1:BRI1-GFP;35S:BAK1-HA* grown in the light condition were put in darkness for 4 days, and treated with different concentrations of glucose for 4 or 24 h. Total proteins were isolated with the extraction buffer and GFP-Trap-A agarose beads were used in the subsequent Co-IP assays. The immunoblots were performed using anti-HA (Abmart M20003, 1/500) and anti-GFP (Abmart M20004, 1/5000) antibodies, respectively. To quantify the interactions between BRI1-GFP and BAK1-HA in response to different concentrations of glucose for 4 h, we measured the intensity of immunoprecipitated BAK1-HA bands and input BAK1-HA bands on blots using ImageJ software. The ratio of immunoprecipitated BAK1-HA to input BAK1-HA was calculated. The ratio value of immunoprecipitated BAK1-HA to input BAK1-HA for 0% glucose treatment was set at 1. Values for glucose treatments are given as mean ± standard deviation (SD) relative to the value for 0% glucose treatment. Three biological replicates were tested.

To investigate effects of glucose on the phosphorylation levels of BRI1, 5-day-old seedlings of *pBRI1:BRI1-GFP* were grown in the light condition were put in darkness for 4 days and treated with different concentrations of glucose for 4 or 24 h. Total proteins from these seedlings were isolated with the extraction buffer, and then immunoprecipitated using GFP-Trap A agarose beads. The eluted proteins were checked using anti-phosphoserine (Sigma P3430, 1/200) and anti-GFP (Abmart M20004, 1/5000) antibodies, respectively. To quantify phosphorylation levels of BRI1 in response to different concentrations of glucose, we measured the intensity of phosphorylated BRI1-GFP bands on Serine residues and immunoprecipitated BRI1-GFP bands on blots using ImageJ software. The ratio of phosphorylated BRI1-GFP on Serine residues to immunoprecipitated BRI1-GFP was calculated. The ratio value of phosphorylated BRI1-GFP on Serine residues to immunoprecipitated BRI1-GFP for 0% glucose treatment was set at 1. Values are given as mean ± standard deviation (SD) relative to the value for 0% glucose treatment. Three biological replicates were tested.

To examine effects of glucose on the phosphorylation levels of BAK1, *pBAK1:BAK1-GFP* seedlings were grown in the light condition for 5 days, and then incubated in darkness for 4 days, followed with different concentrations of glucose for 4 or 24 h. Total proteins from theses seedlings were isolated with the extraction buffer, and then incubated with GFP-Trap-A beads (Chromotek) to immunoprecipitate BAK1-GFP protein. The immunoprecipitated proteins were then examined using anti-phosphothreonine (Cell Signaling 9381, 1/500) and anti-GFP (Abmart M20004, 1/5000) antibodies, respectively. To quantify the phosphorylation levels of BAK1-GFP in response to different concentrations of glucose, we measured the intensity of the phosphorylated BAK1-GFP bands on threonine residues and immunoprecipitated BAK1-GFP bands on blots using ImageJ software. The ratio of phosphorylated BAK1-GFP on threonine residues to immunoprecipitated BAK1-GFP was calculated. The ratio value of phosphorylated BAK1-GFP to immunoprecipitated BAK1-GFP for 0% glucose treatment was set at 1. Values for different glucose treatments are given as mean ± standard deviation (SD) relative to the ratio value for 0% glucose treatment. Three biological replicates were tested.

To investigate effects of BRI1 on the interactions between GPA1 and AGB1, *35S:GPA1-Myc* and *35S:AGB1-HA* plasmids were co-transformed into wild type, *bri1-301,* and *bak1-4* leaf protoplasts. Leaf protoplasts with *35S:AGB1-Myc* and *35S:GPA1-HA* plasmids were grown in darkness for 14 h, and then treated without or with 2% glucose for 5 h. Total proteins from these leaf protoplasts were extracted and immunoprecipitated with anti-Myc-Tag mouse mAb conjugated agarose beads (Abmart M20013), and examined using anti-HA (Abmart M20003, 1/5000) and anti-Myc (Abmart M20002, 1/5000) antibodies, respectively. To quantify the interactions between GPA1 and AGB1 in wild type, *bri1-301,* and *bak1-4* backgrounds, we measure the intensity of immunoprecipitated GPA1-HA bands and input GPA1-HA bands on blots using ImageJ software. The ratio of

immunoprecipitated GPA1-HA to input GPA1-HA was calculated. The ratio value of immunoprecipitated GPA1-HA to input GPA1-HA in response to 0% glucose was set at 1. Values for 2% glucose treatments are given as mean ± SD relative to the ratio value for 0% glucose treatment, set at 1. Three biological replicates were tested.

**Detection of phosphopeptides of AGB1 and AGG3 by LC–MS/MS.** To identify the phosphorylation sites in AGB1, 1 µg GST-BRI1-KD or GST-bri-301-KD proteins were incubated with 2 mg MBP-AGB1 in 100 µl kinase reaction buffer at 30 °C for 3 h. After the reaction, the SDS loading buffer was used to stop the kinase reactions. The phosphorylated proteins were then separated using SDS-PAGE, and electrophoresis was performed at 90 Volts for 2 h. Target bands for MBP-AGB1 were cutted off from the electrophoresis gel and digested using trypsin (Sigma) at 37 °C overnight. LC–MS/MS were then used to analyze the tryptic peptides. Subsequently, these peptides were identified by searching the UniProt database. Phosphosites assignment was performed with the Proteome Discoverer software (version1.3) (Thermo Fisher). Similarly, the phosphorylation sites in AGG3 were identified.

**Constructs and plant transformation**. The *pBAK1;BAK1-GFP* constructs were generated with a PCR-based infusion clone system (GBI GB2001-24). A total of 2100-bp promoter of the *BAK1* gene was obtained with primers gBAK1-GFP-pro-F/R, and 1989-bp CDS of the *BAK1* gene was obtained with primers gBAK1-GFP-CDS-F/R (Supplemental Table 3). Then the mixture of these two PCR products was used as template, and the full-length pBAK1-BAK1 DNA fragment was obtained with primers gBAK1-GFP-pro-F and gBAK1-GFP-CDS-R. By infusion clone system (GBI GB2001-24), pBAK1-BAK1 DNA fragment was cloned into the *pMDC107* plasmid to make a *pBAK1;BAK1-GFP* construct. The *pBAK1;BAK1-GFP* construct was introduced into wild-type seedlings, and transgenic lines were selected for further research[66–69].

**Data availability**. The authors declare that all data supporting the findings of this study are available within the manuscript and the Supplementary Files or are available from the corresponding authors upon request.

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

## Acknowledgements

We thank Cyril Zipfel, Jia Li, Jiayang Li, Yonghong Wang, Wenqiang Tang, and Steve Clouse for BR-related mutant and transgenic seeds and the *Arabidopsis* Stock center NASC for the *bak1-4* mutant. This work was supported by grants from the National Natural Science Foundation of China (31425004, 31571499, 91417304, 31400249, and 91017014), the Ministry of Agriculture of China (2016ZX08009-003) and the Strategic Priority Research Program "Molecular Mechanism of Plant Growth and Development" of CAS (XDPB0401).

## Author contributions

Y.P., L.C., S.L., L.Z., and Y.L. designed research. Y.P., L.C., S.L., Y.Z., Z.L., R.X., W.L., J.K., and L.Z. performed most experiments. X.H. and Y.W. performed mass spectrometry analyses. Y.P., L.C., Y.W., B.C., L.Z., and Y.L. analyzed data. Y.P., L.Z., and Y.L. wrote the article. All the authors have discussed the results and commented the manuscript.

## Additional information

**Competing interests:** The authors declare no competing interests.

