## [Peer Review File · Nature Communications]

Reviewers' comments:

Reviewer #1 (Remarks to the Author):

The manuscript "The brassinosteroid receptor BRI1 and its co-receptor BAK1 interact with G proteins to regulate sugar-responsive growth and development in *Arabidopsis thaliana*" reports on a potential molecular link between the BR and the sugar signaling pathways. The interactions between sugar and BR signaling have been established in previous work and seems to happen molecularly at the level of BIN2 (a glycogen synthase that acts as a negative regulator of the BR pathway) and as well at the level of the master transcription factor BZR1 (that transduce BR signals on the expression of hundreds of genes) in various *Arabidopsis* organs such as the root and the hypocotyl.

For example, here is a non-exhaustive list of previous work that could help evaluate the significance of the work presented here:

a- Zhang et al., *Current Biology* 2016 "TOR Signaling Promotes Accumulation of BZR1 to Balance Growth with Carbon Availability in *Arabidopsis*."

b- Zhang et al., *Planta* 2015. "Brassinosteroid is required for sugar promotion of hypocotyl elongation in *Arabidopsis* in darkness"

c-Gupta et al., *Plant Physiology* 2015. Multiple Interactions between Glucose and Brassinosteroid Signal Transduction Pathways in *Arabidopsis* Are Uncovered by Whole-Genome Transcriptional Profiling.

d-Gupta et al., *Plant Physiology* 2015 Interaction between Glucose and Brassinosteroid during regulation of lateral root development in *Arabidopsis thaliana*

Overall I think that the work is significant since it suggest that sugar homeostasis/signaling is integrated at the level of the BR receptor/co-receptor pair. The topic of the work is significant enough to warrant serious consideration by the editorial board at Nature Communications. Yet the physiological relevance of the work as well as the mechanistic and molecular foundation of the regulation of the BR pathway by sugar (or sucrose in this case) needs to be explored in depth to meet the standards of the journal.

The main data and related comments for the presented work are:

1- BRI1 and BAK1 are required for sugar-responsive growth and development in the dark.

At this point of the work, one wonders if the effects are specific to BRI1 and BAK1 or are rather more general to BR signaling. *det2* or *cpd* mutant plants should have been evaluated in the assay. Conversely a gain-of-function approach would have been appropriate and highly informative. In this respect 35S:BRI1 plants could have been used. Also it would have been nice to assay the effect of sucrose on wildtype plants growing on a media supplemented with either Brassinolide (BL) or Brassinazole (BRZ-a chemical inhibitor of BR biosynthesis)

2- Glucose promotes the interactions and phosphorylation between BRI1 and BAK1.

* I found interesting the fact that glucose can promote the interaction and phosphorylation between BRI1 and BAK1 in a dose dependent manner. Even more interesting is the fact that when supplied at 4% or 6% the promoting effect of Glc on the BAK1-BRI1 interaction and phosphorylation is canceled.

One wonders therefore if wildtype plants growing in the dark with 4 or 6% of glucose still reach the developmental stage 3 and 4 (see Fig.1). If the promotion of the BRI1-BAK1 interaction is causal to the phenotypes depicted in Fig.1 then one would expect that at this concentration of Glc wildtype plants remain at the developmental stage 1. If the interaction is not causal to the phenotype then there is a major mechanistic flaw in the way the story is presented. Also it would have been interesting to test if in the absence of BRs (through inhibition of biosynthesis with BRZ) Glc still promote the BAK1-BRI1 interaction.

* The co-IPs and phosphorylation assays were performed on plants overexpressing BAK1. BAK1 overexpression has been shown to trigger cell death: Domínguez-Ferreras et al., Plant Physiology 2015. Thus one wonders if the assays are skewed in these transgenic plants. Anti-BAK1 and anti-BRI1 antibodies are commercially available. Therefore, the assays could have been performed in truly native conditions and this would have made the story of higher quality.

* To really show that the effect of Glc is really specific to the BRI1-BAK1 receptor co-receptor pair it would have been key to test the interaction between FLS2 and BAK1. Since the latter pair is involved in defense rather than growth the authors could have made a nice point for specificity and exclude 'wide-ranging' unspecific effects of Glc on the plasma-membrane/cell surface. Again FLS2 antibodies are commercially available or could be obtained from the Zipfel or Robatzek labs. I'm fully aware that the authors performed some 'specificity' controls with Mannitol in their assays. However one note that Mannitol impedes Arabidopsis development in both light and dark growing environments.

3- BRI1 and BAK1 act in a common pathway with G protein subunits to control sugar responses

4- BRI1 and BAK1 interact with G protein subunits known to be involved in transducing homeostatic sugar signals

5- BRI1 is required for the interactions between the G protein subunits GPA1 and AGB1

* The wording of this figure legend is in my opinion not appropriate. In fact, in the absence of BRI1, GPA1 and AGB1 interact constitutively in the absence or presence of Glc. In the presence of BRI1 the association between the two G protein subunits in the presence of Glc is lowered.

* The interaction of GPA1 and AGB1 should have been tested in bak1 mutant plants tested in parallel to bri1 plants. bak1 mutant plants phenocopies bri1 plants for glc-dependent growth promotion in the dark.

* Finally, the effects of BRs addition (+BL) or BRs removal (BRZ) on the GPA1-AGB1 interaction should have been evaluated carefully. This would have perhaps provided the desperately missing physiological framework of the story.

All in all, this manuscript reports on another significant interaction between BRs and Glc. Addition of in-depth and well-controlled physiological and biochemical assays would be key to warrant an eventual reconsideration at Nature Comms.

Reviewer #2 (Remarks to the Author):

Brassinosteroid (BR) receptors modulation of G-proteins to regulate sugar signaling pathways. Though there have been several reports in the literature linking G-proteins and BR in regulating early hypocotyl development/seed germination, this work, through powerful and convincing experimental results, establishes a functional connection between BR receptors and G-proteins in sugar signaling. The strength of the manuscript comes from the choice of experimental methods that support the tenet of the manuscript. Specifically use of genetic methods of double mutants in various combinations allowed the manuscript to support the conclusions.

However, to implicate BR and sugar regulation of 'growth' through G-protein, it was expected that the authors would have treated the plants with both BR and sugars. Without any data on the combined effect, it is hard to envision the physiological basis of crosstalk between the BR and sugar pathways. In fact, there are few papers published recently that also draw a connection between BR and sugar pathways- (Gupta et al., *Plant Physiol.* 2015 Jul; 168(3): 1091–1105.; (*Plant Physiol.* 2015 May;168(1):307-20). Though the authors have cited these two papers, but a lack of discussion on the almost opposite findings in these papers is noted. Especially when the findings are almost opposite of what this manuscript proposes. Using genetic studies, it has been shown that BR and sugar pathway crosstalk only in the HKK-dependent pathway; whereas HKK-independent pathway that involves the G-proteins and RGS, the BR and sugar pathway do not crosstalk as all the mutants in the G-protein pathway showed the same BR responses as found in the wild type plants. Moreover, the use of BR in concert with the sugar, clearly showed that it is the HKK dependent pathway that is used to mediate the BR and sugar crosstalk not the G-protein dependent sugar pathway.

The manuscript was not able to discuss how mechanistically the glucose can function at the BR receptor level to activate the G-protein mediated signaling. The discussion section is very poorly written as it appeared as the repetition of the Results section. There was not much discussion about the implication/mechanism of function that can be deduced from the results. A simple Model figure for the proposed pathway would have helped in strengthening the Discussion section.

There are many different alleles of the mutants are available and a justifications for the use of the particular alleles were warranted. Choice of the alleles which are different from the alleles used in other papers that drew opposite results and these discussions should have been included in the discussion to 'refute' the BR and sugar crosstalk ONLY in the HKK-dependent pathway not in the G-protein pathway.

When BRI1 is required to have GPA1 and AGB1 interact in response to glucose (fig. 6), it is hard to envision how GPA1 is not involved in the BR and sugar cross-talk while AGB1 is involved. Detailed explanation is required in the discussion (undetectable level of interaction of BRI1 with GPA1 may not explain the conclusion). In the face of number of interactions of sugars with diverse hormones, yet another interactions with Brassinosteroid calls for a working model figure to facilitate the proper placement of this reported interactions in the sugar pathways.

Many of the experiments were done with single mutants of BRI1 or BAK1; hence failed to show whether the effects are seen is due the single gene or whether presence of both of BRI1 and BAK1 is required for the effect. Why not overexpress BAK1 in bri-101 background or BRI1 in bak1 background to eliminate this possibility or pBRI1:BRI1-GFP;35S:BAK1-HA plants could have been combined with G-protein mutants.

Suggestions are: use the gin2 mutants to 'refute' or 'support' the published link between the BR and sugar pathways; use both BR and sugar together to implicate a crosstalk between these two growth

regulators; and consider the fact that both BRI1 and BAK1 together can mediate the crosstalk (so over-expression of only a single receptor may not stoichiometrically function); consider the published possibility of "Glc and BR act antagonistically at low Glc concentration and synergistically at higher Glc concentrations for hypocotyl elongation growth regulation in dark-grown seedlings." (Upto 2% of glucose BRI1 and BAK1 interacts but at higher concentration- no interaction- Fig 2)

Reviewer #3 (Remarks to the Author):

The manuscript describes that BRI1 and BAK1 interact and phosphorylate AGB1 and AGG3 in response to sugar treatment. Sugar responses in single mutants and epistasis analysis supports the idea that BRI1/BAK1 work through G proteins to regulate sugar responses in parallel to RGS1.

The manuscript contains interesting information and most data are convincing; however, in order to show that the model that the authors proposed has biological meaning, the authors should identify the phosphorylation sites of AGB1 and AGG3 and transform the phospho-dead or -mimic variants back into the mutant background to see if sugar signaling is affected. In the two recent studies by Choudhury and Pandey (plant Cell 2015, 27: 3260) and Liang et al., (ELife,2016, 5:e13568), they addressed the phosphorylation-dependent regulation of G-proteins by RLKs, and importantly, both studies confirmed the biological meaning by using phospho-dead and/or -mimic mutants. Therefore although this study contains interesting information, its mechanism is not as thoroughly addressed as previously published studies.

Another concern is the time points used for analysis (4h-24h after sugar treatment) are very long compared to usual receptor interaction or phosphorylation experiments- I am concerned that some of the effects could therefore be indirect, downstream consequences of the treatments.

In addition, some minor concerns:

1. Both BRI1 and BAK1 are serine/threonine kinases, why the authors only use the anti-pThr antibody to detect the threonine phosphorylation of AGB1 and AGG3? Did the authors also try to use the anti-pSer antibody to detect the serine phosphorylation of both proteins?
2. In Line 236-239, the authors claimed 'However, neither *agg1* and *agg2* single mutants or *agg1 agg2* double mutants exhibited the small organ phenotype, suggesting that AGG3 plays a major role in plant growth and development. Therefore, we focused on testing the genetic interactions of BRI1 and BAK1 with AGG3 in sugar-responsive growth.' Although *agg1 agg2* double mutants do not show strong developmental phenotype, we cannot conclude that they do not participate in BRI1-BAK1 mediated sugar signaling. So, the authors could test if BRI1 and BAK1 also interact and phosphorylate other AGGs.
3. In Figure 2, the authors should use BAK1-HA expressing lines without BRI-GFP, as a negative control for the Co-IP experiment. In addition, the authors could test by microscopy if BRI1 and BAK1 were internalized upon the glucose treatment. If so, the mechanisms would be different from that the authors proposed.
4. In Figure 4 A and B, the input lanes of MBP-GPA1 are dirty. So, it is difficult to tell which bands indicate MBP-GPA1. If the authors cannot find the good MBP antibodies, they should at least use arrows to point which band is MBP-GPA1.
5. In Figure 5 C-F, the quality of western blot was not very good, especially for Figure 5D. In this

case, it would be more reliable to do replicates and show statistics as shown in Figure 2. As aforementioned, did the authors try to use the anti-pSer antibody to detect the serine-phosphorylation of AGB1 and AGG3?

6. In Figure 6A, the IP: a-Myc, IB: a-HA bands for bri1-301 and all the inputs bands for a-Myc are over-exposed. It is not accurate to do the quantification using over-exposed bands. In addition, for bri1-301, seems more GPA1-HA was pulled down by AGB1-Myc after being treated with 2% glucose. But it could be because there was more GPA1-HA input. The major issue is the top bands were over-exposed, thus it is hard to tell if the interaction between GPA1 and AGB1 was affected by glucose in bri mutants.

Dear Reviewers

We are very grateful to the reviewers for your helpful and supportive suggestions on the manuscript. We have carefully taken reviewers' comments into consideration in preparing this revision. We have performed experiments suggested by three reviewers and addressed reviewers' concerns. They have helped make a better paper.

Reviewer #1

The manuscript "The brassinosteroid receptor BRI1 and its co-receptor BAK1 interact with G proteins to regulate sugar-responsive growth and development in *Arabidopsis thaliana*" reports on a potential molecular link between the BR and the sugar signaling pathways. The interactions between sugar and BR signaling have been established in previous work and seems to happen molecularly at the level of BIN2 (a glycogen synthase that acts as a negative regulator of the BR pathway) and as well at the level of the master transcription factor BZR1 (that transduce BR signals on the expression of hundreds of genes) in various *Arabidopsis* organs such as the root and the hypocotyl.

For example, here is a non-exhaustive list of previous work that could help evaluate the significance of the work presented here:

- a- Zhang et al., *Current Biology* 2016 "TOR Signaling Promotes Accumulation of BZR1 to Balance Growth with Carbon Availability in *Arabidopsis*."
- b- Zhang et al., *Planta* 2015. "Brassinosteroid is required for sugar promotion of hypocotyl elongation in *Arabidopsis* in darkness"
- c-Gupta et al., *Plant Physiology* 2015. Multiple Interactions between Glucose and Brassinosteroid Signal Transduction Pathways in *Arabidopsis* Are Uncovered by Whole-Genome Transcriptional Profiling.
- d-Gupta et al., *Plant Physiology* 2015 Interaction between Glucose and Brassinosteroid during regulation of lateral root development in *Arabidopsis thaliana*

Overall I think that the work is significant since it suggest that sugar homeostasis/signaling is integrated at the level of the BR receptor/co-receptor pair.

The topic of the work is significant enough to warrant serious consideration by the editorial board at Nature Communications. Yet the physiological relevance of the work as well as the mechanistic and molecular foundation of the regulation of the BR pathway by sugar (or sucrose in this case) needs to be explored in depth to meet the standards of the journal.

ANSWER: Thank you very much for your very help and positive suggestions. We performed experiments as suggested by the reviewer1 (see below).

The main data and related comments for the presented work are:

1- BRI1 and BAK1 are required for sugar-responsive growth and development in the dark.

At this point of the work, one wonders if the effects are specific to BRI1 and BAK1 or are rather more general to BR signaling. *det2* or *cpd* mutant plants should have been evaluated in the assay. Conversely a gain-of-function approach would have been appropriate and highly informative. In this respect *35S:BRI1* plants could have been used. Also it would have been nice to assay the effect of sucrose on wild type plants growing on a media supplemented with either Brassinolide (BL) or Brassinazole (BRZ-a chemical inhibitor of BR biosynthesis)

ANSWER: As suggested by the reviewer1, we quantified the developmental stages of *det2* and *35S:BRI1* seedlings grown in the dark. 53% of wild-type seedlings grown on medium with 1% glucose developed to stage 4, while only 12% of *det2* mutants developed to stage 4, suggesting that a proper level of BR is required for sugar responses (Supplementary Fig.20). In contrast, 64% of *35S:BRI1* seedlings had developed to stage 4 (Supplementary Fig.20).

As suggested by the reviewer1, we investigated the effects of Brassinolide (BL) and

Brassinazole (BRZ, a chemical inhibitor of BR biosynthesis) on sugar-responsive dark development of wild-type seedlings. When seedlings were grown on medium with glucose or sucrose, BRZ strongly inhibited sugar-responsive dark development. There was no obvious difference between the BL-treated seedlings and the control when seedlings were grown on medium with glucose or sucrose. However, the inhibitory effects of BRZ on dark development could be rescued by BL (Supplementary Fig. 8). These results suggest that a proper level of BR is required for sugar-responsive growth and development.

2- Glucose promotes the interactions and phosphorylation between BRI1 and BAK1.

* I found interesting the fact that glucose can promote the interaction and phosphorylation between BRI1 and BAK1 in a dose dependent manner. Even more interesting is the fact that when supplied at 4% or 6% the promoting effect of Glc on the BAK1-BRI1 interaction and phosphorylation is canceled. One wonders therefore if wildtype plants growing in the dark with 4 or 6% of glucose still reach the developmental stage 3 and 4 (see Fig.1). If the promotion of the BRI1-BAK1 interaction is causal to the phenotypes depicted in Fig.1 then one would expect that at this concentration of Glc wild type plants remain at the developmental stage 1. If the interaction is not causal to the phenotype then there is a major mechanistic flaw in the way the story is presented.

ANSWER: As suggested by the reviewer1, we quantified the developmental stages of wild-type seedlings grown on medium with high glucose in the dark. Most of wild-type seedlings grown on medium with 1% glucose had developed to stage 4, while most of wild-type seedlings grown on medium with 4% or 6% glucose developed to stage 1 - stage 3 (see the following figure). In addition, in dark-grown Col-0 seedlings, hypocotyl length increased in response to low concentrations of glucose, and elongation was progressively inhibited at high concentrations of glucose (Supplementary Fig. 2).

These results indicate that the interactions between BRI1 and BAK1 are correlated with sugar responses.

The effects of different concentrations of glucose or mannitol on dark development of wild-type seedlings.

Comparison of developmental stages of wild-type seedlings grown on medium with different concentrations of glucose (G) and mannitol (M) as indicated in the dark for 19 days ($n \geq 78$). Values are given as mean \pm standard deviation (SD).

Also it would have been interesting to test if in the absence of BRs (through inhibition of biosynthesis with BRZ) Glc still promote the BAK1-BRI1 interaction.

ANSWER: As suggested by the reviewer1, we investigated whether BRZ influences the promoting effect of glucose on the interactions between BRI1 and BAK1 *in vivo*. As shown in Supplementary Fig. 22a, the BRI1 and BAK1 interactions induced by 2% glucose were obviously decreased when BRZ was added for 6 hours. In addition, the BRI1 and BAK1 interactions were almost abolished when we extended the BRZ treatment time (Supplementary Fig. 22b). These results suggest that a proper level of BR is required for the BRI1 and BAK1 interactions induced by glucose in plants.

* The co-IPs and phosphorylation assays were performed on plants overexpressing BAK1. BAK1 overexpression has been shown to trigger cell death: Domínguez-Ferreras et al., Plant Physiology 2015. Thus one wonders if the assays are skewed in these transgenic plants. Anti-BAK1 and anti-BRI1 antibodies are

commercially available. Therefore, the assays could have been performed in truly native conditions and this would have made the story of higher quality.

ANSWER: As suggested by the reviewer1, we examined effects of glucose on interactions and phosphorylations of BRI1 and BAK1 using native antibodies. As shown in Supplementary Fig. 3, interactions of BRI1 and BAK1 were progressively increased in response to 1% and 2% glucose, and then gradually decreased in response to 4% and 6% glucose, consistent with the results using tag-specific antibodies. Similarly, the phosphorylation levels of BRI1 and BAK1 were increased in response to 1% and 2% glucose, and then started to decrease in response to 4% and 6% glucose (Supplementary Fig.5), consistent with our results using tag-specific antibodies. Thus, these results further confirm that glucose influences the interactions and the phosphorylation levels of BRI1 and BAK1 in a concentration-dependent manner.

* To really show that the effect of Glc is really specific to the BRI1-BAK1 receptor co-receptor pair it would have been key to test the interaction between FLS2 and BAK1. Since the latter pair is involved in defense rather than growth the authors could have made a nice point for specificity and exclude 'wide-ranging' unspecific effects of Glc on the plasma-membrane/cell surface. Again FLS2 antibodies are commercially available or could be obtained from the Zipfel or Robatzek labs. I'm fully aware that the authors performed some 'specificity' controls with Mannitol in their assays. However one notes that Mannitol impedes Arabidopsis development in both light and dark growing environments.

ANSWER: As suggested by the reviewer1, we examined whether glucose could effects the interaction between FLS2 and BAK1. As shown in Supplementary Fig.21, interactions between FLS2 and BAK1 are strongly increased when treated with flg22, while interactions between FLS2 and BAK1 did not show obvious difference when treated with or without 2% glucose, indicating that glucose does not affect the interaction of FLS2 and BAK1.

3- BRI1 and BAK1 act in a common pathway with G protein subunits to control sugar responses

4- BRI1 and BAK1 interact with G protein subunits known to be involved in transducing homeostatic sugar signals

5- BRI1 is required for the interactions between the G protein subunits GPA1 and AGB1

* The wording of this figure legend is in my opinion not appropriate. In fact, in the absence of BRI1, GPA1 and AGB1 interact constitutively in the absence or presence of Glc. In the presence of BRI1 the association between the two G protein subunits in the presence of Glc is lowered.

ANSWER: As suggested by the reviewer1, we reworded this sentence.

* The interaction of GPA1 and AGB1 should have been tested in bak1 mutant plants tested in parallel to bri1 plants. bak1 mutant plants phenocopies bri1 plants for glc-dependent growth promotion in the dark.

ANSWER: As suggested by the reviewer1, we tested the interaction of GPA1 and AGB1 in *bak1-4* mutants. As shown in Fig. 7, the interactions between GPA1 and AGB1 in wild-type protoplasts were strongly decreased in response to exogenous glucose. On the contrary, their interactions in *bak1-4* protoplasts were only slightly decreased in response to exogenous glucose. These results indicated that BAK1 also influences the interactions of AGB1 and GPA1 in response to glucose.

* Finally, the effects of BRs addition (+BL) or BRs removal (BRZ) on the GPA1-AGB1 interaction should have been evaluated carefully. This would have perhaps provided the desperately missing physiological frame work of the story.

ANSWER: As suggested by the reviewer1, we tested the effects of BL and BRZ on the interaction between GPA1 and AGB1. As shown in Supplementary Fig. 25, BL did not obviously affect the interactions between GPA1 and AGB1, while BRZ strongly increased their interactions, consistent with the effects of BL and BRZ on sugar-regulated dark development (Supplementary Fig. 9). These results indicate that a proper level of BR is required for sugar-regulated G-protein interactions.

All in all, this manuscript reports on another significant interaction between BRs and Glc. Addition of in-depth and well-controlled physiological and biochemical assays would be key to warrant an eventual reconsideration at Nature Comms.

Reviewer #2 (Remarks to the Author):

Brassinosteroid (BR) receptors modulation of G-proteins to regulate sugar signaling pathways. Though there have been several reports in the literature linking G-proteins and BR in regulating early hypocotyl development/seed germination, this work, through powerful and convincing experimental results, establishes a functional connection between BR receptors and G-proteins in sugar signaling. The strength of the manuscript comes from the choice of experimental methods that support the tenet of the manuscript. Specifically use of genetic methods of double mutants in various combinations allowed the manuscript to support the conclusions.

However, to implicate BR and sugar regulation of 'growth' through G-protein, it was expected that the authors would have treated the plants with both BR and sugars. Without any data on the combined effect, it is hard to envision the physiological basis of crosstalk between the BR and sugar pathways. In fact, there are few papers published recently that also draw a connection between BR and sugar pathways- (Gupta et al., Plant Physiol. 2015 Jul; 168(3): 1091–1105.; (Plant Physiol. 2015 May;168(1):307-20).

Though the authors have cited these two papers, but a lack of discussion on the almost opposite findings in these papers is noted. Especially when the findings are almost opposite of what this manuscript proposes. Using genetic studies, it has been shown that BR and sugar pathway crosstalk only in the HKK-dependent pathway; whereas HKK-independent pathway that involves the G-proteins and RGS, the BR and sugar pathway do not crosstalk as all the mutants in the G-protein pathway showed the same BR responses as found in the wild type plants. Moreover, the use of BR in concert with the sugar, clearly showed that it is the HKK- dependent pathway that is used to mediate the BR and sugar crosstalk not the G-protein dependent sugar pathway.

ANSWER: As suggested by the reviewer2, we analyzed the combined effects of BL, BRZ and glucose on G protein mutants. The dark development of the wild type was inhibited by BRZ (Supplementary Fig.9). This inhibition of dark development was partially rescued with the addition of BL. By contrast, the dark development of *gpa1-101*, *agb1-2*, and *agg3-3* was insensitive to the inhibition of BRZ compared with that of the wild type (Supplementary Fig.9). Interestingly, *gpa1-101*, *agb1-2* or *agg3-3* hypocotyls exhibited similar responses to wild-type hypocotyls when treated with BRZ or BRZ/BL (Supplementary Fig. 10). Consistent with this, previous studies showed that BR and sugar crosstalk acts through HXK1 pathway to regulate hypocotyl elongation in plants (Gupta et al., Plant Physiology, 2015, 168: 1091- 1105). Under the light condition, *gpa1-101*, *agb1-2*, and *agg3-3* seedlings were more insensitive to BRZ when seedlings were grown on medium with 6% glucose (Supplementary Fig. 11). These results indicate that the crosstalks between BR and sugar can act through G protein subunits in an organ-dependent manner. Similarly, BR and auxin crosstalk acts through IAA proteins to regulate downstream factors in an organ dependent manner in *Arabidopsis* (Nakamura et al., Plant Journal, 2006, 45, 193–205). In addition, BR and auxin crosstalk also acts through *OsIAA* to modulate plant growth in an organ-dependent manner in rice (Song et al., Plant Mol Biol, 2009, 70:297–309).

The manuscript was not able to discuss how mechanistically the glucose can function at the BR receptor level to activate the G-protein mediated signaling. The discussion section is very poorly written as it appeared as the repetition of the Results section. There was not much discussion about the implication/mechanism of function that can be deduced from the results. A simple Model figure for the proposed pathway would have helped in strengthening the Discussion section.

ANSWER: As suggested by the reviewer2, we rewrote the discussion and made a simple model for the proposed pathway (Supplementary Fig. 26).

There are many different alleles of the mutants are available and a justifications for the use of the particular alleles were warranted. Choice of the alleles which are different from the alleles used in other papers that drew opposite results and these discussions should have been included in the discussion to 'refute' the BR and sugar crosstalk ONLY in the HKK-dependent pathway not in the G-protein pathway.

ANSWER: In this revision, we found that the G-protein pathway is required for BR and sugar crosstalks in regulating dark development and high glucose-induced developmental arrest (Supplementary Fig. 9 and 11). However, hypocotyls of G protein mutants showed similar response to those of the wild type (Supplementary Fig. 10). Consistent with this, previous studies showed that BR and sugar crosstalk acts through HXK1 pathway to regulate hypocotyl elongation in plants (Gupta et al., Plant Physiology, 2015, 168: 1091-1105). These results indicate that the crosstalks between BR and sugar can act through G protein subunits in an organ-dependent manner.

The *bri1-301* mutation leads to loss of kinase activity, which is widely studied (Noguchiet al., Plant Physiology, 1999, 121:743-752; Xu et al., Cell Research, 2008, 18: 472-478). The *bri1-301* is in Col-0 background, and G protein mutants are also in Col-0 background. It is convenient to do genetic analysis. *bak1-4* is a null allele of *BAK1* gene, and *bak1-4* mutants show similar growth phenotypes with *bak1-1* (Li et al., Cell,

2002, 110: 213–222; He et al., *Current Biology*, 2007, 17: 1109–1115; Ntoukakis et al., *The Plant Cell*, 2011, 23: 3871–3878). *gpa1-101*, *agb1-2* and *agg3-3* were also described in previous studies (Gupta et al., *Plant Physiology*, 2015, 168: 1091-1105; Li et al., *New Phytologist*, 2012, 194: 690–703; Ullah et al., *Science*, 2001, 292: 2066-2069).

When BRI1 is required to have GPA1 and AGB1 interact in response to glucose (fig. 6), it is hard to envision how GPA1 is not involved in the BR and sugar cross-talk while AGB1 is involved. Detailed explanation is required in the discussion (undetectable level of interaction of BRI1 with GPA1 may not explain the conclusion). In the face of number of interactions of sugars with diverse hormones, yet another interactions with Brassinosteroid calls for a working model figure to facilitate the proper placement of this reported interactions in the sugar pathways.

ANSWER: Our genetic analyses showed that BRI1 and BAK1 act partially through GPA1 in response to glucose, although BRI1 and BAK1 do not physically interact with GPA1 (Fig. 3). Thus, GPA1 is genetically involved in BR and sugar cross-talks. BRI1 and BAK1 influence the interactions of GPA1 with AGB1 in response to glucose (Fig. 7). Thus, BRI1 and BAK1 act through G-protein subunits (GPA1, AGB1 and AGGs) in response to glucose by directly targeting AGB1 and AGGs, not GPA1.

As suggested by the reviewer2, we made a simple working model for this proposed pathway (Supplementary Fig.26).

Many of the experiments were done with single mutants of BRI1 or BAK1; hence failed to show whether the effects are seen is due the single gene or whether presence of both of BRI1 and BAK1 is required for the effect. Why not overexpress BAK1 in *bri-101* background or BRI1 in *bak1* background to eliminate this possibility or pBRI1:BRI1-GFP;35S:BAK1-HA plants could have been combined with G-protein mutants.

ANSWER: As suggested by the reviewer2, we examined the effects of overexpression of *BRI1* in *bak1* or *BAK1* in *bri1*, respectively. As shown in the following figure a, overexpression of *BRI1* in *bak1-4* or overexpression of *BAK1* in *bri1-301* slightly complement the corresponding mutant defects in response to glucose. It is possible reason that overexpression of *BRI1* or *BAK1* causes sugar sensitive phenotypes, therefore slight complementation. These results indicate that *BRI1* and *BAK1* are simultaneously required for sugar-responsive growth.

As suggested by the reviewer2, we examined the effects of both *BRI1* and *BAK1* overexpression in *agb1*, *gpa1* and *agg3* mutants, respectively. As shown in the following figure b, overexpression of both *BRI1* and *BAK1* in wild-type seedlings decreased the green seedlings ratio. However, overexpression of both *BRI1* and *BAK1* in *agb1*, *gpa1* and *agg3* mutants, respectively, did not obviously decrease the green seedlings ratio of these mutants. Thus, these results suggest that *BRI1* and *BAK1* act through G-proteins in response to glucose.

Sugar responses of the indicated seedlings overexpressing *BRI1* and *BAK1*.

(a) Comparison of development stages of the indicated seedlings grown on 1% glucose (G) medium in the dark for 19 days ($n \geq 54$).

(b) The percentage of the indicated seedlings with green cotyledons. Seedlings were grown on 6% glucose medium under constant light condition for 15 days ($n \geq 60$).

Values (a,b) are given as mean \pm standard deviation (SD). $**P < 0.01$ compared with the wild type by Student's *t*-test.

Suggestions are: use the *gin2* mutants to 'refute' or 'support' the published link between the BR and sugar pathways; use both BR and sugar together to implicate a crosstalk between these two growth regulators; and consider the fact that both BRI1 and BAK1 together can mediate the crosstalk (so over-expression of only a single receptor may not stoichiometrically function); consider the published possibility of "Glc and BR act antagonistically at low Glc concentration and synergistically at higher Glc concentrations for hypocotyl elongation growth regulation in dark-grown seedlings. " (Upto 2% of glucose BRI1 and BAK1 interacts but at higher concentration- no interaction- Fig 2)

ANSWER: Our results show that the G-protein pathway is required for BR and sugar crosstalk in regulating dark development and high glucose-induced developmental arrest (Supplementary Fig.9 and 11). However, hypocotyls of G protein mutants showed similar response to those of the wild type (Supplementary Fig.10), consistent with the previous reports (Gupta et al., *Plant Physiology*, 2015, 168: 1091-1105). Previous studies showed that BR and sugar crosstalk acts through HXK1 pathway to regulate hypocotyl elongation in plants (Gupta et al., *Plant Physiology*, 2015, 168: 1091-1105). These results suggest that BR and sugar crosstalk acts through G-proteins in an organ-dependent manner. Similarly, BR and auxin crosstalk acts through IAA proteins to regulate downstream factors in an organ dependent manner in *Arabidopsis* (Nakamura et al., *Plant Journal*, 2006, 45, 193–205). In addition, BR and auxin crosstalk also acts through *OsIAA* to modulate plant growth in an organ-dependent manner in rice (Song et al., *Plant Mol Biol*, 2009, 70:297–309). We discussed this in this revision.

One paper reported that Glc and BR act antagonistically at low Glc concentration and synergistically at higher Glc concentrations for hypocotyl elongation growth regulation in dark-grown seedlings (Gupta et al., *Plant Physiology*, 2015, 168:

1091-1105). However, another study showed that the promoting effect of sugar (90 mM glucose) on hypocotyl elongation was not affected by BL treatment, but decreased by BRZ treatment in a concentration-dependent manner. The inhibition effect of BRZ can be partially rescued by BL (Zhang et al., *Planta*, 2015, 242(4):881-93), suggesting that BL could enhance hypocotyl elongation at low sugar concentration. These results indicate that glucose and BL did not act antagonistically at low Glc concentration. Our results are consistent with this study (Zhang et al., *Planta*, 2015, 242(4):881-93).

Reviewer #3 (Remarks to the Author):

The manuscript describes that BRI1 and BAK1 interact and phosphorylate AGB1 and AGG3 in response to sugar treatment. Sugar responses in single mutants and epistasis analysis supports the idea that BRI1/BAK1 work through G proteins to regulate sugar responses in parallel to RGS1.

The manuscript contains interesting information and most data are convincing; however, in order to show that the model that the authors proposed has biological meaning, the authors should identify the phosphorylation sites of AGB1 and AGG3 and transform the phospho-dead or -mimic variants back into the mutant background to see if sugar signaling is affected. In the two recent studies by Choudhury and Pandey (*Plant Cell* 2015, 27: 3260) and Liang et al., (*ELife*, 2016, 5:e13568), they addressed the phosphorylation-dependent regulation of G-proteins by RLKs, and importantly, both studies confirmed the biological meaning by using phospho-dead and/or -mimic mutants. Therefore although this study contains interesting information, its mechanism is not as thoroughly addressed as previously published studies.

ANSWER: As suggested by the reviewer3, we identified the phosphorylation sites of AGB1 and AGG3, and transformed phospho-dead and phospho-mimicking constructs of *AGB1* and *AGG3* into *agb1-2* and *agg3-3* mutants, respectively. As shown in Fig. 6c,

the green cotyledon percentages of transgenic lines overexpressing AGB1 or phospho-mimicking forms of AGB1 in *agb1-2* were similar to those of the wild type when they were grown on medium with 6% glucose, while the green cotyledon percentages of transgenic lines overexpressing AGB1^{T14A,S40A,T243A,T253A} in *agb1-2* were significantly lower than those of the wild type. These results indicate that the phosphorylation of amino acids T14, S40, T243, and T253 in AGB1 is important for sugar responses in *Arabidopsis*. Similarly, the phosphorylation of amino acids S21D, S22D, T143D and T199D in AGG3 is important for sugar responses in *Arabidopsis* (Fig. 6d).

Another concern is the time points used for analysis (4h-24h after sugar treatment) are very long compared to usual receptor interaction or phosphorylation experiments- I am concerned that some of the effects could therefore be indirect, downstream consequences of the treatments.

ANSWER: As suggested by the reviewer3, we examined the effects of glucose on interactions and phosphorylations of BRI1 and BAK1 in short time treatment. As shown in Supplementary Fig. 23, interactions and phosphorylations of BRI1 and BAK1 were obviously increased when we treated with glucose for a short time (0.5 h).

In addition, some minor concerns:

1. Both BRI1 and BAK1 are serine/threonine kinases, why the authors only use the anti-pThr antibody to detect the threonine phosphorylation of AGB1 and AGG3? Did the authors also try to use the anti-pSer antibody to detect the serine phosphorylation of both proteins?

ANSWER: As suggested by the reviewer3, we also use the anti-phosphoserine (pSer) antibody to detect the serine phosphorylation of AGB1 and AGG3. As shown in Supplementary Fig. 18, serine phosphorylation levels of AGB1 and AGG3 were also

obviously increased in seedlings overexpressing *BRI1*, but reduced in *bri1-301* seedlings.

2. In Line 236-239, the authors claimed ‘However, neither *agg1* and *agg2* single mutants or *agg1 agg2* double mutants exhibited the small organ phenotype, suggesting that *AGG3* plays a major role in plant growth and development. Therefore, we focused on testing the genetic interactions of *BRI1* and *BAK1* with *AGG3* in sugar-responsive growth.’ Although *agg1 agg2* double mutants do not show strong developmental phenotype, we cannot conclude that they do not participate in *BRI1-BAK1* mediated sugar signaling. So, the authors could test if *BRI1* and *BAK1* also interact and phosphorylate other *AGGs*.

ANSWER: As suggested by the reviewer³, we tested if *BRI1* and *BAK1* could interact and phosphorylate other *AGGs*. As shown in Supplementary Fig. 14a,c, *BRI1* can interact with *AGG1* and *AGG2* *in vitro* and *in vivo*. *BAK1* can also interact with *AGG1* and *AGG2* *in vitro* and *in vivo* (Supplementary Fig. 14b,d). We then investigated whether *BRI1* and *BAK1* can phosphorylate these *AGGs*. As shown in Supplementary Fig. 15, *AGG1* and *AGG2* were phosphorylated by *BRI1* kinase domain (*BRI1-KD*) using *in vitro* phosphorylation assays. Similarly, *AGG1*, *AGG2* and *AGG3* were phosphorylated by *BAK1* kinase domain (*BAK1-KD*) (Supplementary Fig. 16).

3. In Figure 2, the authors should use *BAK1-HA* expressing lines without *BRI-GFP*, as a negative control for the Co-IP experiment. In addition, the authors could test by microscopy if *BRI1* and *BAK1* were internalized upon the glucose treatment. If so, the mechanisms would be different from that the authors proposed.

ANSWER: As suggested by the reviewer³, we used *BAK1-HA* lines as a negative control for Co-IP experiment in this revision (Fig. 2a).

As suggested by the reviewer³, we also examined the plasma membrane localization of *BRI1-GFP* and *BAK1-GFP* using confocal microscope. Recent studies have shown

that BR signaling is mainly activated by the plasma-membrane localized BRI1 (Martins et al., *Nature Communications*, 2015, 6:6151|DOI: 10.1038/ncomms7151; Irani et al., *Nature Chemical Biology*, 2012, 8: 583-589). BR signaling is enhanced when BRI1 internalization is inhibited, and BRI1 endocytosis is mainly required for BR signal attenuation (Irani et al., *Nature Chemical Biology*, 2012, 8: 583-589; Martins et al., *Nature Communications*, 2015, 6:6151|DOI: 10.1038/ncomms7151; Rubbo et al., *The Plant Cell*, 2013, 25: 2986–2997). Considering that glucose affects the interactions and phosphorylations of BRI1 and BAK1, we investigated whether glucose could affect the plasma membrane localizations of BRI1 and BAK1 in plants. The plasma membrane localizations of BRI1-GFP and BAK1-GFP were not strongly affected in response to glucose from 0% to 2%, consistent with the increased BR signaling induced by low glucose (Supplementary Fig. 7a-d). However, the plasma membrane localizations of BRI1-GFP and BAK1-GFP were significantly decreased in response to glucose from 4% to 6% (Supplementary Fig. 7a,b), consistent with the notion that high concentrations of glucose inhibit plant growth. Thus, these results indicate that glucose modulates the plasma membrane localizations of BRI1 and BAK1 in a concentration-dependent manner.

4. In Figure 4 A and B, the input lanes of MBP-GPA1 are dirty. So, it is difficult to tell which bands indicate MBP-GPA1. If the authors cannot find the good MBP antibodies, they should at least use arrows to point which band is MBP-GPA1.

ANSWER: As suggested by the reviewer³, we used arrows to point which band is MBP-GPA1 in this revision (Fig. 4a,b).

5. In Figure 5 C-F, the quality of western blot was not very good, especially for Figure 5D. In this case, it would be more reliable to do replicates and show statistics as shown in Figure 2. As aforementioned, did the authors try to use the anti-pSer antibody to detect the serine-phosphorylation of AGB1 and AGG3?

ANSWER: As suggested by the reviewer3, we quantified the phosphothreonine level of AGB1 and AGG3 in corresponding plants and showed the statistics in this revision (Fig. 5c-j).

As suggested by the reviewer3, we had used the anti-phosphoserine (pSer) antibodies to detect the serine phosphorylation of AGB1 and AGG3 in this revision. As shown in Supplementary Fig. 18, serine phosphorylation levels of AGB1 and AGG3 were also obviously increased in seedlings overexpressing *BRI1*, but reduced in *bri1-301* seedlings.

6. In Figure 6A, the IP: a-Myc, IB: a-HA bands for *bri1-301* and all the inputs bands for a-Myc are over-exposed. It is not accurate to do the quantification using over-exposed bands. In addition, for *bri1-301*, seems more GPA1-HA was pulled down by AGB1-Myc after being treated with 2% glucose. But it could be because there was more GPA1-HA input. The major issue is the top bands were over-exposed, thus it is hard to tell if the interaction between GPA1 and AGB1 was affected by glucose in *bri* mutants.

ANSWER: As suggested by the reviewer3, we repeated these experiments for several times. We used new images in this revision (Fig. 7).

Reviewers' comments:

Reviewer #2 (Remarks to the Author):

The answers to the 1st round of comments appear satisfactory

Reviewer #3 (Remarks to the Author):

The authors have addressed all of my comments in their revised manuscript and I now consider it ready for publication.

Reviewer #4 (Remarks to the Author):

This manuscript (NCOMMS-16-29072A) reports on a new mechanism for crosstalk between the phytohormone BR and sugar signaling in regulating seedling development in the dark, which was shown to be mediated by the interactions between the BR receptors BRI1/BAK1 and the G proteins in the sugar response pathways. This finding is definitely interesting because both BR and sugars regulate many different developmental processes in plants and the identification of different molecular links between the BR and sugar signaling pathways will help explain how BR and sugars coordinately regulate different developmental processes.

Since this is a revised manuscript and the previous reviewers have raised very good questions regarding the mechanisms as to how BRI1/BAK1 interact with G-proteins. To avoid extra work and endless questions to the authors, I will only comment on some of the existing issues that were asked by previous reviewers (particularly reviewer 1) but not sufficiently or adequately addressed by the authors.

1. The reviewer 1 has raised a very good question that at the high concentrations of glucose (4 and 6%), the interaction between BRI1 and BAK1 was abolished and the reviewer was wondering whether this abolishment is correlated with the seedling development phenotype (i.e. whether the seedlings can still develop into stages 3 and 4, or remain at stage 1, at these high concentrations). The authors performed a new experiment on this and found that at these high concentrations, some of the plants can still develop into stages 3 and 4, only small portion of seedlings remains at stage 1. These results suggest that the interaction between BRI1 and BAK1 are substantial mediators of BR- and sugar-regulated seedling development but not the entire mechanisms. Therefore, the authors need to re-tune their statement that BRI1 and BAK1 interaction is required for BR- and sugar-regulated seedling development. The current results suggest that BRI1 and BAK1 interaction is part of the mechanisms of BR- and sugar-regulated seedling development, but not required.
2. The second issue is that the authors' genetic data suggests that BR1/BAK1 work through G proteins to regulate sugar responses in parallel to RGS1. This result raises an immediate question that whether BR1/BAK1 are physically interacting with RGS1 and therefore can modify RGS1 activity. If this is the case, it will add another piece of evidence to support the idea that BR and sugar regulate plant development through the HXK1-independent pathways.
3. From the results that G-protein mutants *gpa1-101*, *agb1-2* or *agg3-3* show similar responses to wild-type plants in hypocotyl elongation when treated with BRZ or BRZ/BL under both dark and light conditions, the authors concluded that the crosstalk between BR and sugar acts through G protein subunits in an organ-dependent manner. I do not agree with the authors that this result suggests that BR and sugar acts through G protein subunits in an organ-dependent manner, instead, it suggests that BR and sugar regulate different aspects or processes of seedling development through different

mechanisms (considering the different results of the previous studies). I suggest the authors to modify their conclusions on this point. In addition, the authors need to define what specific processes of dark seedling development are regulated by BRI1/BAK1 and G proteins (e.g. leaf-like structure formation or internode elongation?). Dark seedling development is too general to describe the specific processes regulated by these factors.

Minor points:

4. Lines 119-121, for classification of seedling development: The sentence "dark-grown Arabidopsis seedlings grown on MS medium with glucose were classified into four different developmental stages" is grammatically not correct. It should be "the development of dark-grown Arabidopsis seedlings grown on MS medium with glucose were classified into four different developmental stages".

5. Figure 1 a: What was the plant material or genotype used for experiment here? Wild type? This should be indicated in the figure legend.

6. Figure 1 b & c: Any image phenotypes of the plants can be shown for the plants tested? That will be more straight forward for readers to understand how important that BRI1 and BAK1 are to sugar-regulated plant development in the dark.

Dear Reviewers

We are very grateful to the reviewers for your helpful and positive comments on this manuscript. The reviewer4 gives very helpful suggestions, and other reviewers have no further concerns. We have carefully taken reviewer4's comments and the editor's suggestions into consideration in preparing this revision. We have performed experiments suggested by the reviewer4 and addressed reviewer4' concerns. They have helped to make a better paper. We would like to submit this revision for possible publication in Nature Communications.

Responses to Reviewers1/2/3

ANSWER: Thank you very much for your very positive comments.

Responses to Reviewer #4

This manuscript (NCOMMS-16-29072A) reports on a new mechanism for crosstalk between the phytohormone BR and sugar signaling in regulating seedling development in the dark, which was shown to be mediated by the interactions between the BR receptors BRI1/BAK1 and the G proteins in the sugar response pathways. This finding is definitely interesting because both BR and sugars regulate many different developmental processes in plants and the identification of different molecular links between the BR and sugar signaling pathways will help explain how BR and sugars coordinately regulate different developmental processes.

Since this is a revised manuscript and the previous reviewers have raised very good questions regarding the mechanisms as to how BRI1/BAK1 interact with G-proteins. To avoid extra work and endless questions to the authors, I will only comment on some of the existing issues that were asked by previous reviewers (particularly reviewer 1) but not sufficiently or adequately addressed by the authors.

ANSWER: Thank you very much for your very helpful and positive suggestions. We performed experiments as suggested by you (see below).

1. The reviewer 1 has raised a very good question that at the high concentrations of glucose (4 and 6%), the interaction between BRI1 and BAK1 was abolished and the reviewer was wondering whether this abolishment is correlated with the seedling development phenotype (i.e. whether the seedlings can still develop into stages 3 and 4, or remain at stage 1, at these high concentrations). The authors performed a new experiment on this and found that at these high concentrations, some of the plants can still development into stages 3 and 4, only small portion of seedlings remains at stage 1. These results suggest that the interaction between BRI1 and BAK1 are substantial mediators of BR- and sugar-regulated seedling development but not the entire mechanisms. Therefore, the authors need to re-tune their statement that BRI1 and BAK1 interaction is required for BR- and sugar-regulated seedling development. The current results suggest that BRI1 and BAK1 interaction is part of the mechanisms of BR- and sugar-regulated seedling development, but not required.

ANSWER: As suggested by the reviewer #4, we have modified our previous statement in this revision.

2. The second issue is that the authors' genetic data suggests that BR1/BAK1 work through G proteins to regulate sugar responses in parallel to RGS1. This result raises an immediate question that whether BR1/BAK1 are physically interacting with RGS1 and therefore can modify RGS1 activity. If this is the case, it will add another piece of evidence to support the idea that BR and sugar regulate plant development through the HXK1-independent pathways.

ANSWER: As suggested by the reviewer #4, we tested whether BRI1 and BAK1 interact with RGS1. As shown in Supplementary Fig. 26a-c, our results indicated that BRI1 and BAK1 interact with RGS1 *in vitro* and *in vivo*, respectively.

3. From the results that G-protein mutants *gpa1-101*, *agb1-2* or *agg3-3* show similar responses to wild-type plants in hypocotyl elongation when treated with BRZ or

BRZ/BL under both dark and light conditions, the authors concluded that the crosstalk between BR and sugar acts through G protein subunits in an organ-dependent manner. I do not agree with the authors that this result suggests that BR and sugar acts through G protein subunits in an organ-dependent manner, instead, it suggests that BR and sugar regulate different aspects or processes of seedling development through different mechanisms (considering the different results of the previous studies). I suggest the authors to modify their conclusions on this point. In addition, the authors need to define what specific processes of dark seedling development are regulated by BRI1/BAK1 and G proteins (e.g. leaf-like structure formation or internode elongation?). Dark seedling development is too general to describe the specific processes regulated by these factors.

ANSWER: As suggested by the reviewer #4, we have modified our previous conclusion in this version. As suggested by the reviewer #4, BR and sugar may regulate different processes of seedling development through different mechanisms. In addition, as suggested by the reviewer #4, we defined the processes of dark seedling development regulated by BRI1/BAK1 and G proteins in this version. BRI1/BAK1 and G proteins play important roles in sugar-responsive processes including leaf-like structure formation, cotyledon expansion and internode elongation.

Minor points:

4. Lines 119-121, for classification of seedling development: The sentence “dark-grown Arabidopsis seedlings grown on MS medium with glucose were classified into four different developmental stages” is grammatically not correct. It should be “the development of dark-grown Arabidopsis seedlings grown on MS medium with glucose were classified into four different developmental stages”.

ANSWER: As suggested by the reviewer #4, we corrected the grammatical error in this version.

5. Figure 1 a: What was the plant material or genotype used for experiment here? Wild type? This should be indicated in the figure legend.

ANSWER: As suggested by the reviewer #4, we indicated the plant material used in this experiment is wild type (Col-0).

6. Figure 1 b & c: Any image phenotypes of the plants can be shown for the plants tested? That will be more straight forward for readers to understand how important that BRI1 and BAK1 are to sugar-regulated plant development in the dark.

ANSWER: As suggested by the reviewer #4, we showed the representative phenotypes used for the data statistics of Figure 1 b & c. And image phenotypes are shown in the Supplementary Fig. 1.

REVIEWERS' COMMENTS:

Reviewer #4 (Remarks to the Author):

The authors have addressed all my questions. I have no further concerns.